# Play to Generalize:
# Learning to Reason Through Game Play

**Yunfei Xie[1], Yinsong Ma[2], Shiyi Lan[3], Alan Yuille[2], Junfei Xiao[2],[*] Chen Wei[1][†]**
[1]Rice University,    [2]Johns Hopkins University,    [3]NVIDIA

## Abstract

Developing reasoning capabilities in multimodal large language models (MLLMs) remains challenging. Motivated by literature suggesting that gameplay promotes transferable reasoning skills, we propose a novel post-training method, Visual Game Learning (ViGaL), where MLLMs develop generalizable reasoning skills through playing arcade-like games. Specifically, we show that training a 7B-parameter MLLM via reinforcement learning (RL) on simple games like Snake significantly enhances the downstream performance on multimodal math benchmarks like MathVista, and on multi-discipline questions like MMMU, without seeing any worked solutions, equations, or diagrams during RL. Remarkably, our model outperforms specialist models post-trained on benchmark-oriented multimodal reasoning data, while preserving the model's performance on general visual benchmarks, a challenge where specialist models often fall short. Our findings suggest that multimodal reasoning can emerge from gameplay, pointing to a promising strategy of designing surrogate tasks for RL post-training. The code is available at `https://yunfeixie233.github.io/ViGaL`.

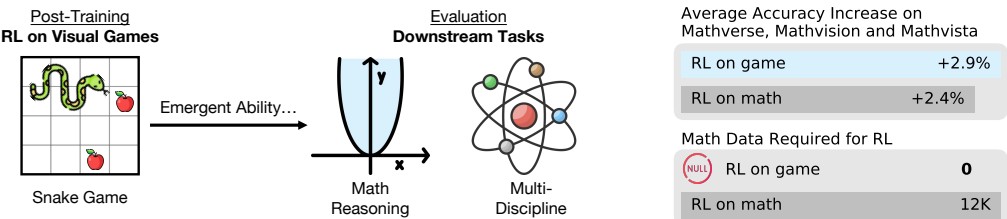

Figure 1: **Overview of ViGaL.** *Left*: We propose a novel post-training method where MLLMs are finetuned via RL to play arcade-style games such as Snake (Kamradt, 2025). We demonstrate that gameplay post-training enables MLLMs to achieve out-of-domain generalization, enhancing their performance on downstream multimodal reasoning tasks requiring math, spatial and multi-discipline reasoning, without using math or multi-displine data during RL. *Right*: Our ViGaL (RL on game) achieves higher average accuracy increase than MM-Eureka (Meng et al., 2025) (RL on math) across three multimodal math benchmarks. This is notable because MM-Eureka trains on large-scale, curated math datasets, while ViGaL only uses game data. Details are in Tab 2.

## 1 Introduction

Games, beyond their entertainment value, provide rich and diverse structured environments for developing and studying general reasoning and problem-solving abilities. Humans from early childhood acquire foundational cognitive skills through diverse game-like activities such as arranging objects, navigating spaces, and manipulating tools. These experiences foster essential building blocks of abstract thinking, including pattern recognition, spatial reasoning, and causal inference (Brändle et al., 2021; Bertram, 2020). In cognitive science, games are used as experimental platforms to reveal the inductive biases of the human mind (Allen et al., 2024; Alhasoun & Alneghiemish, 2021), such as planning depth in the game Four-in-a-Row (Van Opheusden et al., 2023), or the cognitive basis of tool use through the game Virtual Tools (Allen et al., 2020).

AI agents, too, have benefited from games resembling aspects of human play. These environments encourage exploration, robustness to sparse rewards, and learning from multimodal inputs. For example, emergent tool use has been observed in agents trained via hide-and-seek (Baker et al., 2019), and Atari gameplay has been incorporated into training generalist agents (Reed et al., 2022). By learning in these environments, AI systems develop robust and transferable reasoning capabilities.

Recent work has shown that post-training with Reinforcement Learning (RL) can unlock reasoning behaviors from their base models (DeepSeek-AI, 2025; OpenAI, 2024). These RL-trained models can "think before they speak", generating internal chain-of-thought traces before outputting a final answer. More importantly, growing evidence suggests that RL often generalizes more robustly to out-of-distribution samples than supervised fine-tuning (SFT). For example, models trained with RL on mathematics transfer their reasoning skills to physics (Meng et al., 2025), and navigation agents adapt to novel environments beyond their training domains (Chu et al., 2025a). Motivated by these findings, we ask: *since games already serve as a natural medium through which humans acquire reasoning strategies, can post-training multimodal LLMs on gameplay similarly enhances their ability to reason across diverse tasks?*

The results are striking (Fig. 1). We show that post-training a 7B-parameter multimodal LLM, Qwen2.5-VL-7B (Bai et al., 2023), to play simple arcade-style games like Snake (Kamradt, 2025) yields two surprising outcomes: (1) the model generalizes to previously unseen games (Sec. 2.3); and (2) it exhibits strong reasoning abilities on multimodal math benchmarks like MathVista (Lu et al., 2024), and multi-domain QA like MMMU (Yue et al., 2024a). Despite never observing worked solutions, equations, or diagrams during RL post-training, the model achieves competitive results not only against large-scale industrial systems like GPT-4o (Hurst et al., 2024), but also against specialist models post-trained on math datasets (Tabs. 2 and 3). Furthermore, it improves on reasoning benchmarks without degrading general visual understanding, a common limitation of domain-specialist training (Tab. 9). Overall, gameplay emerges as an effective surrogate task for incentivizing reasoning in multimodal LLMs.

Why does it work? Our ablation studies suggest that reasoning skills incentivized by gameplay can be helpful to other multimodal reasoning tasks. For example, Snake, a game set on a *2D grid* where the player maneuvers the "snake" to avoid collisions and collect apples, significantly improves performance on math problems involving *2D coordinates*. In contrast, Rotation, a puzzle requiring recognition of 3D *object rotation angles*, more strongly boosts performance on *geometry questions involving angles and lengths* (Fig. 3). Furthermore, jointly training on both games yields consistently stronger results on downstream benchmarks than training on either game alone, suggesting the **compositionality** of the acquired skills (Tab. 2).

All these results point to a new post-training strategy: rather than relying solely on domain-specific datasets, we can design scalable and controllable **surrogate tasks for post-training**, such as games, that unlock reasoning behaviors transferable to downstream applications. Synthetic game environments offer structured, rule-based rewards and fine-grained controllability, while also scaling far more easily than human-annotated data. This promising paradigm of post-training with surrogate tasks reminisces self-supervised pre-training in vision and language (He et al., 2020; Doersch et al., 2015; Radford et al., 2018), where carefully designed pretext tasks produce broad generalization.

## 2 REINFORCEMENT LEARNING ON VISUAL GAMES

We introduce ViGaL, a novel post-training paradigm designed to enhance generalization capabilities.

### 2.1 GAME ENVIRONMENT

As show in Fig. 2, under our ViGaL paradigm, the model is trained in a game environment where it receives states from game environment, outputs next actions, and obtains rewards as feedback from the environment. Formally, each task, given an instruction $I$, can be formulated as a partially observable Markov decision process (POMDP): $(\mathcal{S}, \mathcal{A}, \mathcal{O}, T, R, \Omega)$, where $\mathcal{S}$ is the set of possible environment states, $\mathcal{O}$ is the set of observations available to the model, and $\mathcal{A}$ represents actions model can do in this game environment. $T : \mathcal{S} \times \mathcal{A} \rightarrow \mathcal{S}$ is the state transition function, while $R$ is a binary reward from the environment representing the correctness of action. Due to partial observability, the agent perceives only observations $o = \Omega(s)$.

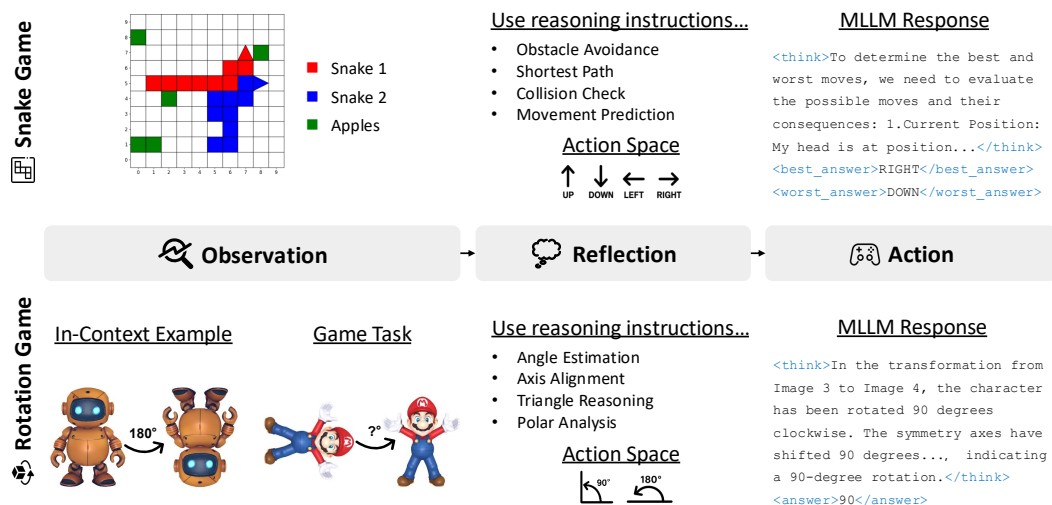

Figure 2: **Post-training MLLMs to reason through RL with games.** We propose post-training MLLMs via RL by playing visual games. We demonstrate this with two games: the classic arcade game Snake (Kamradt, 2025), and Rotation, a self-designed task to investigate spatial reasoning. In each game, the model receives multimodal inputs and follows reasoning instructions, *e.g.*, path planning in Snake, angle estimation in Rotation. It reflects to choose an action, outputs its chain-of-thoughts and decision, *e.g.*, best/worst move or predicted angle, and receives a reward. Through gameplay, the model obtains reasoning abilities that transfer to downstream multimodal reasoning tasks such as math and multi-discipline question answering.

**Snake and Rotation Games.**    We design two complementary games, Snake and Rotation, to study the proposed paradigm (Fig. 2), each focusing on different MLLM capabilities. The Snake game, inspired by prior work showing that competition can enhance reasoning in MLLMs (Du et al., 2023), emphasizes strategic decision-making. We set up a dual-snake game based on SnakeBench (Kamradt, 2025), where each model independently controls one snake. The objective is to reach apples, score points, and outcompete the opponent. At time $t$, the environment state $s^t$ includes the coordinates of both snakes $(x_{s_i}^t, y_{s_i}^t)$ for $i \in \{1, 2\}$, the apple location $(x_a^t, y_a^t)$, and the previous actions $A_i^{t-1}$. All elements are placed on a $10 \times 10$ board. Each snake then selects its next action $A_i^t \in \{\texttt{up}, \texttt{down}, \texttt{left}, \texttt{right}\}$. A snake dies if it collides with itself, the other snake, or the board boundary; the survivor wins, or in the case of simultaneous death, the higher score decides. Unlike SnakeBench, which uses only text to represent states, we provide both images of the game board and textual descriptions as observations $o^t = \Omega(s^t)$ for richer input. The Rotation game, inspired by rotation-angle prediction as a pre-text task in self-supervised learning (Gidaris et al., 2018), evaluates visual perception and spatial reasoning. The model is presented with two views of the same 3D object: an initial view $I_{\text{init}}$ and a rotated view $I_{\text{rot}}$, obtained by rotating the object $90°$ or $180°$ around the $z$-axis (pointing toward the viewer). The task is to identify which rotation angle transforms $I_{\text{init}}$ into $I_{\text{rot}}$. To guide reasoning, we include an in-context example with a known rotation. As in the Snake game, observations combine images and text. Together, these two games allow systematic exploration of reasoning and perception, two fundamental aspects of MLLM abilities.

## 2.2    RULE-BASED REINFORCEMENT LEARNING

We apply rule-based RL to directly post-train MLLMs for visual games, without relying on supervised learning as a warm up. The algorithm is described as follows:

**Reward design.**    We use a simple rule-based reward function to avoid reward hacking (Gao et al., 2022) and help the model learn how to play the games effectively. This reward function has two components: an accuracy reward and a format reward. The total reward $r$ is computed as the sum of an accuracy reward and a format reward $r = r_{\text{accuracy}} + r_{\text{format}}$. The accuracy reward $r_{\text{accuracy}}$ is 1 if the answer is correct, and 0 otherwise. Details of reward for each game are in Appendix Sec. A.3.

| Model | Wins (/10) | Model | Acc. (%) |
|---|---|---|---|
| ViGaL vs. | | ViGaL | **71.9** |
| Qwen2.5-VL-7B | 9 | Qwen2.5-VL-7B | 47.4 |
| Qwen2.5-VL-72B | 7 | Qwen2.5-VL-72B | 52.1 |
| Llama-4-Maverick | 7 | Llama-4-Maverick | 66.2 |
| Gemini-2.5-Pro | 8 | Gemini-2.5-Pro | 51.0 |
| Claude-3.7-Sonnet | 6 | Claude-3.7-Sonnet | 65.6 |
| GPT-4o | 8 | GPT-4o | 61.5 |
| o4-mini | 6 | o4-mini | 70.8 |

| Game | ViGaL | Qwen2.5-VL-7B |
|---|---|---|
| Space Invaders | 280.0 | 85.0 |
| Ms. Pacman | 1370.0 | 670.0 |
| Seaquest | 80.0 | 60.0 |
| Alien | 540.0 | 450.0 |
| Frogger | 7.0 | 5.0 |
| Breakout | 0.0 | 9.0 |
| Pong | -26.0 | -26.0 |
| Cumulative Reward | **2251.0** | 1253.0 |

(a) Snake game.  (b) Rotation game.  (c) Atari game.

Table 1: **Game Performance.** (a) ViGaL gets high win rates (6-9 wins out of 10 matches) on Snake playing against advanced proprietary models. (b) ViGaL shows best performance on Rotation. (c) ViGaL trained on Snake and Rotation shows zero-shot generalization to unseen Atari games, achieving a nearly *doubled* cumulative reward compared to its base model (Qwen2.5-VL-7B).

**Advantage estimation and policy update.** We employ REINFORCE Leave-One-Out (RLOO) algorithm (Kool et al., 2019; Ahmadian et al., 2024) in our RL training phase. Following Group Policy Gradient (Chu et al., 2025b), we omit KL divergence regularization. Without KL constraints limiting policy changes, the model explores the solution space more freely, potentially discovering better reasoning strategies. This enables more flexible adaptation during RL training.

### 2.3 IMPLEMENTATION AND EVALUATION ON GAMES

**Implementation details.** We employ Qwen2.5-VL-7B-Instruct (Bai et al., 2023) as our base model. We follow DeepSeek-R1 (DeepSeek-AI, 2025), using a combination of rule-based format rewards and accuracy rewards, with RLOO (Kool et al., 2019; Ahmadian et al., 2024) as the core RL algorithm. We implement our training within a multimodal input RL framework based on Open-RLHF (Hu et al., 2024). For hyperparameters, we adopt the default settings from MM-Eureka (Meng et al., 2025), including a global batch size of 128, a rollout batch size of 128, a rollout temperature of 1.0, and a learning rate of $1e^{-6}$. Training uses 6 A100-80G GPUs.

**Game training data.** We build game environments to collect training data for our experiments. For Snake, we leverage SnakeBench (Kamradt, 2025) as our data engine. For Rotation, we utilize Hunyuan3D (Team, 2025), which generates 3D meshes from images or text instructions. We render each mesh into 2D images from different orientations, creating image pairs with associated rotation angles as ground truth labels for RL training. Our comprehensive data generation pipeline enables producing training samples at any desired scale with fully customized settings. For experiments, we synthesize 36K samples per game, sufficient for convergence. Details are in Appendix Sec. A.1.

**Competing with leading models on Snake and Rotation.** To evaluate the game capabilities of ViGaL models, we initialize environments in diverse states unseen during training. For Snake (Tab. 1a), we randomly initialize games 10 times with two models competing directly, measuring win counts. For Rotation (Tab. 1b), we measure rotation angle prediction accuracy on comprehensive validation sets with 3D object meshes unseen during training. Our 7B-parameter model consistently outperforms proprietary models in both games. Results confirm that RL effectively unlocks small 7B models' ability to excel in visual games requiring environmental understanding, reasoning, planning, and interactive decision-making.

**Out-of-distribution generalization to Atari games.** We then test ViGaL on Atari-GPT (Waytowich et al., 2024), a benchmark for evaluating MLLMs as decision-making agents in Atari video games such as in Fig. 5. The benchmark consists of seven different Atari games, with detailed settings in Appendix Sec. B.1. We follow most settings and prompts from Atari-GPT, with a small modification to ensure format correctness for all models. Following Atari-GPT (Waytowich et al., 2024), we report cumulative reward over 1K steps as the evaluation metric, where higher rewards indicate better performance. As shown in Tab. 1c, ViGaL shows significant cumulative reward improvement on Atari games despite being trained only on Snake and Rotation games. This is particularly notable because Atari games differ substantially from our training games in both visual appearance and gameplay strategies. These results suggest that our rule-based RL training approach enables strong generalization to previously unseen game environments.

| Model | Avg. | | Math | | | | Geometry | |
|---|---|---|---|---|---|---|---|---|
| | | Avg. | MathVista | MathVerse | MathVision | Avg. | GeoMath | Geo3K |
| Proprietary Model | | | | | | | | |
| GPT-4o (Hurst et al., 2024) | 47.5 | 47.3 | 61.4 | 50.2 | 30.4 | 46.8 | 50.2 | 43.5 |
| Gemini-2.0-Flash (Team, 2023) | 55.4 | 56.4 | 73.4 | 54.6 | 41.3 | 54.4 | 55.3 | 53.5 |
| Multimodal Reasoning Model Post-Trained on Qwen2.5-VL-7B (Bai et al., 2023) | | | | | | | | |
| *Base Model (Qwen2.5-VL-7B)* | 46.3 | 47.7 | 68.0 | 49.0 | 26.0 | 44.8 | 44.0 | 45.6 |
| R1-Onevision-7B (Yang et al., 2025) | 40.9 | 46.8 | 64.1 | 46.4 | **29.9** | 35.0 | 45.4 | 24.5 |
| R1-VL-7B (Chen et al., 2025b) | 40.9 | 42.7 | 63.5 | 40.0 | 24.7 | 39.0 | 42.0 | 36.1 |
| MM-Eureka-Qwen-7B (Meng et al., 2025) | 39.3 | 50.1 | 73.0 | 50.3 | 26.9 | 28.4 | 53.1 | 3.8 |
| Reason-RFT-Zero-7B (Tan et al., 2025) | 46.5 | 38.1 | 60.7 | 35.3 | 18.3 | 54.9 | 55.0 | 54.8 |
| VLAA-Thinker-7B (Chen et al., 2025a) | 51.3 | 48.7 | 68.0 | 51.7 | 26.4 | 53.9 | 51.1 | 56.6 |
| OpenVLThinker-7B (Deng et al., 2025) | 52.1 | 47.8 | 70.2 | 47.9 | 25.3 | 56.4 | 49.2 | 63.5 |
| ViGaL Snake | 51.6 | 49.4 | 70.7 | 51.1 | 26.5 | 55.0 | 49.9 | 60.0 |
| ViGaL Rotation | 52.8 | 49.3 | 71.2 | 50.4 | 26.3 | **57.9** | 51.7 | 64.1 |
| ViGaL Snake + Rotation | **53.9** | **50.6** | 71.9 | 52.4 | 27.5 | 57.1 | 51.0 | 63.3 |
| | ±0.3 | ±0.3 | ±0.4 | ±0.2 | ±0.3 | ±0.5 | ±0.3 | ±0.4 |

Table 2: **Results on multimodal mathematical benchmarks.** We compare to other multimodal reasoning models. Results post-trained on the same subject as the evaluation are de-emphasized, while our ViGaL models only use games for post-training. **Bold** numbers are the best in each Avg. column. We include standard deviations of three independent runs for ViGaL Snake + Rotation.

| Model | Avg. | | CLEVR+ | | | Multi-Discipline | |
|---|---|---|---|---|---|---|---|
| | | Avg. | CLEVR-M | S-CLEVR | Avg. | MMMU_val | MMMU-Pro_overall |
| Proprietary Model | | | | | | | |
| GPT-4o (Hurst et al., 2024) | 55.9 | 51.2 | 68.1 | 34.3 | 60.5 | 69.1 | 51.9 |
| Gemini-2.0-Flash (Team, 2023) | – | 46.3 | 64.9 | 27.6 | – | 71.9 | – |
| Multimodal Reasoning Model Post-Trained on Qwen2.5-VL-7B (Bai et al., 2023) | | | | | | | |
| *Base Model: Qwen2.5-VL-7B* | 50.3 | 54.9 | 74.6 | 35.2 | 45.7 | 54.3 | 37.0 |
| R1-Onevision-7B (Yang et al., 2025) | 53.7 | 65.1 | 75.5 | 54.7 | 42.3 | 51.9 | 32.6 |
| R1-VL-7B (Chen et al., 2025b) | 53.9 | 68.0 | 87.4 | 48.6 | 39.7 | 50.0 | 29.4 |
| MM-Eureka-Qwen-7B (Meng et al., 2025) | 62.8 | 79.3 | 98.4 | 60.1 | 46.4 | 55.8 | 36.9 |
| Reason-RFT-Zero-7B (Tan et al., 2025) | 58.6 | 76.2 | 99.4 | 53.0 | 40.9 | 51.2 | 30.6 |
| VLAA-Thinker-7B (Chen et al., 2025a) | 61.7 | **83.4** | 94.7 | 72.1 | 40.1 | 48.2 | 31.9 |
| OpenVLThinker-7B (Deng et al., 2025) | 60.4 | 82.4 | 93.8 | 71.0 | 38.5 | 54.8 | 22.1 |
| ViGaL Snake | 64.4 | 82.6 | 92.6 | 72.6 | 46.2 | 55.8 | 36.6 |
| ViGaL Rotation | 63.3 | 80.7 | 93.0 | 68.3 | 45.9 | 54.1 | 37.7 |
| ViGaL Snake + Rotation | **64.7** | 81.7 | 91.9 | 71.4 | **47.7** | 58.0 | 37.4 |

Table 3: **Results on multimodal spatial and multi-discipline reasoning benchmarks.** CLEVR-M denotes CLEVR-Math (Lindström & Abraham, 2022), and S-CLEVR stands for Super-CLEVR (Li et al., 2023). Results post-trained on the same subject as the evaluation are de-emphasized, while ViGaL is exclusively post-trained using games. **Bold** numbers are the best in each Avg. column.

## 3 VISUAL REASONING GENERALIZATION

**Evaluation collection.** Following prior studies (Tong et al., 2024a; Li et al., 2024c), we systematically divide existing benchmarks into two broad categories: (i) *reasoning-oriented benchmarks* requiring multi-step or mathematical reasoning, and (ii) *general-purpose perception benchmarks* assessing visual understanding and perception abilities.

For reasoning-oriented evaluation, we test on four key areas: *Math* (MathVista (Lu et al., 2024), MathVerse (Zhang et al., 2024), MathVision (Wang et al., 2024b)), *Geometry* (GeoMath (Gao et al., 2023; Shi et al., 2024), Geometry3K (Lu et al., 2021)), *CLEVR+* (CLEVR-Math (Lindström & Abraham, 2022), Super-CLEVR (Li et al., 2023)), and *Multi-Discipline* (MMMU (Yue et al., 2024a), MMMU-Pro (Yue et al., 2024b)). For general perception, we evaluate across three cate-

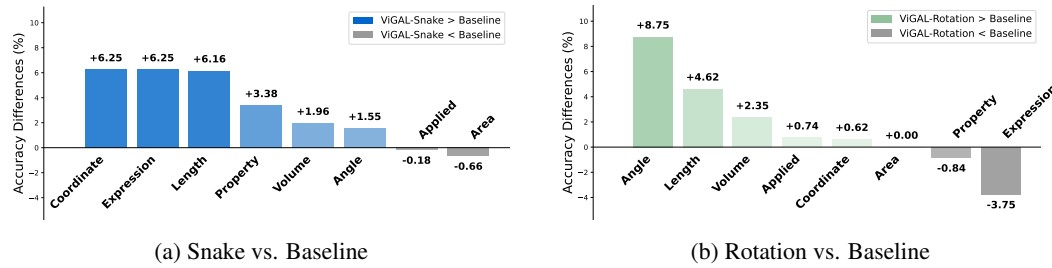

(a) Snake vs. Baseline         (b) Rotation vs. Baseline

Figure 3: **Per-category gains on MathVerse are _not_ uniform.** The eight math categories follow MathVerse (Zhang et al., 2024). (a) Snake yields the largest gains on _Coordinates_ and _Expressions_, consistent with its 2D grid structure. (b) Rotation boosts _Angle_ and _Length_ questions but reduces _Expression_ accuracy, suggesting its training primarily incentivizes orientation recognition.

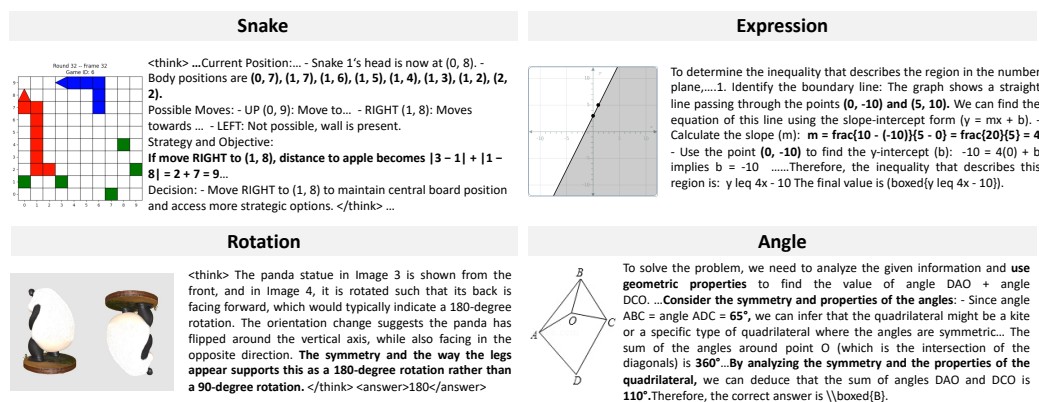

Figure 4: **Reasoning trace of different games and math questions.** Top: Algebraic functions and coordinate-level interpretations that emerge from playing the Snake game help solving _Expression_ questions. Bottom: Spatial reasoning skills incentivized by playing the Rotation game appear when solving _Angle_-related problems.

gories: General (MuirBench (Wang et al., 2024a), CRPE (Kazemzadeh et al., 2014)), Vision-Centric (MMVP (Tong et al., 2024b), RealWorldQA (X.AI, 2024), MMStar (Chen et al., 2024a), MME (Fu et al., 2023), BLINK (Fu et al., 2024)), and OCR & Chart (AI2D (Kembhavi et al., 2016), SEED-Bench-2-Plus (Li et al., 2024a), DocVQA (Mathew et al., 2021), OCRBench (Liu et al., 2023)). More detailed descriptions of each benchmark are provided in Appendix B.6.

## 3.1 MAIN RESULTS

**Zero-shot generalization from gameplay to multimodal reasoning.** Our approach consistently shows remarkable generalization capabilities on mathematical and other reasoning tasks, despite having no direct exposure to in-domain training data during RL post-training. As shown in Tab. 2, our method notably outperforms models specifically RL-trained on mathematical tasks. For instance, ViGaL Snake + Rotation achieves 0.5% higher accuracy than MM-Eureka-Qwen-7B (Meng et al., 2025) 28.7% on Geometry, even though MM-Eureka-Qwen-7B was explicitly trained on high-quality mathematical and geometry datasets.

This strong generalization extends beyond mathematics. Tab. 3 shows that ViGaL Snake + Rotation outperforms R1-OneVision-7B (Yang et al., 2025) by 5.4% on average across MMMU series benchmarks, which test multi-disciplinary reasoning. This is particularly notable since R1-OneVision-7B was trained on a carefully curated comprehensive dataset spanning multiple subjects.

These empirical results suggest that gameplay-based post-training develops fundamental reasoning capabilities that transfer more effectively than direct RL training on diverse task-specific datasets. Moreover, the gameplay environment appears to encourage general problem-solving strategies that consistently generalize well to out-of-domain tasks.

| Model | Avg. | MathVista | MathVerse | MathVision |
|-------|------|-----------|-----------|------------|
| Base Model: Qwen2.5-VL-7B | 47.7 | 68.0 | 49.0 | 26.0 |
| MM-Eureka-Qwen-7B | 50.1 | 73.0 | 50.3 | 26.9 |
| ViGaL Snake + Rotation | 50.6 | 71.9 | 52.4 | 27.5 |
| ViGaL Snake + Rotation + Math Data | 51.8 | 72.3 | 54.5 | 27.7 |

Table 4: **Gameplay complements math data.** Adding math data MMK12 on top of ViGaL yields further gains in math performance. With access to the same amount of math data, ViGaL outperforms MM-Eureka (Meng et al., 2025) on average of the three math benchmarks.

**Blending multiple games enhances generalization.**   As shown in Tab. 2, post-training on Snake achieves best performance on the CLEVR+ benchmark, while training on Rotation yields stronger results on geometry reasoning. Furthermore, training on both Snake and Rotation enables learning complementary skills, improving the overall benchmark average to 63.1%. These findings indicate that combining game environments drives meaningful performance gains, demonstrating Visual Gaming Learning's potential as a promising paradigm for enhancing generalizable reasoning without large-scale domain-specific data. Expanding the types of games during training consistently scales performance across different visual reasoning tasks.

**Different games benefit distinct math subfields.**   To study which types of problems in the math benchmarks benefit from game play, we analyze accuracy differences across MathVerse (Zhang et al., 2024) subcategories between ViGaL models trained with Snake or Rotation, as shown in Fig. 3. We find that training on the Snake game significantly improves performance on the subcategories like Expressions and Coordinates, while training on Rotation notably enhances performance on questions about angles and lengths. To understand why different games help with different types of math, we compare the reasoning processes required for playing games versus solving math problems. As shown in Fig. 4, solving Expressions questions involves algebraic functions and coordinate-level interpretations of graphical representations, which closely align with the spatial reasoning process in Snake. Similarly, solving angle-related questions is consistent with requirement of playing Rotation game to reason about rotational angles of 3D objects. These results suggest that playing different games develops fundamental skills like spatial modeling and algebraic calculation that transfer to visual math questions. To validate whether this pattern generalizes beyond Snake and Rotation, we extend this analysis to four additional games (Maze, Tetris, Sudoku, Sokoban) and identify systematic transfer patterns based on cognitive alignment between game mechanics and mathematical skills. A complete multi-game study using K-Means clustering analysis on MathVerse subject-level categories is presented in Appendix Sec. B.4. The experiment on quantitatively analyzing the correlation between math and game is in Sec. B.9 in the Appendix. Furthermore, joint training on both games leads to improvements across *all* reasoning categories (see Appendix Sec. B.3). We also include qualitative analyses on improvements in math reasoning after RL in Appendix Sec. C.

**Gameplay complements math data.**   We explore the complementary benefits of adding math data to the gameplay training pipeline, for which we implement a two-stage training process. Stage 1 equals to ViGaL setup, training the model on Snake and Rotation games. In stage 2, we further finetune the stage 1 model on MMK12 (Meng et al., 2025), a multimodal mathematical reasoning dataset containing approximately 12k examples. Stage 2 training uses the identical data and settings as MM-Eureka-Qwen-7B (Meng et al., 2025). As shown in Tab. 4, the integration of mathematical data in stage 2 yields a continuous improvement of 1.2% on average across three mathematical benchmarks. This demonstrates the complementary relationship between our visual game learning approach and mathematical data post-training. Moreover, ViGaL with math data significantly outperforms MM-Eureka-Qwen-7B by 1.7% on mathematical benchmarks on average, using the same math data. These results suggest that visual game learning can serve as an effective surrogate task together with domain-specific data to improve performance on target tasks.

**Preserving general visual capabilities while reasoning enhancement.**   To examine whether generalization on reasoning tasks leads to degradation in general visual capabilities, we evaluate ViGaL Snake + Rotation on a broader set of MLLM benchmarks. As shown in Tab. 9 in Appendix, compared to Qwen2.5-VL-7B prior to RL tuning, our model maintains comparable general visual performance while achieving stronger math reasoning results. In contrast, other models that improve

(a) Text prompt design.

| prompt | Avg. | Math | CLEVR+ | Geo. |
|---|---|---|---|---|
| base model | 49.1 | 47.7 | 54.9 | 44.8 |
| w/o reasoning instr. | 59.5 | 48.0 | 80.4 | 50.1 |
| w/ reasoning instr. | **62.3** | 49.4 | 82.6 | 55.0 |

(b) Reward design.

| reward | Avg. | Math | CLEVR+ | Geo. |
|---|---|---|---|---|
| base model | 49.1 | 47.7 | 54.9 | 44.8 |
| best moves | 59.6 | 48.2 | 80.4 | 50.2 |
| best & worst moves | **62.3** | 49.4 | 82.6 | 55.0 |
| w/ random label | 49.4 | 47.5 | 55.4 | 47.5 |

(c) Difficulty control.

| difficulty control | Avg. | Math | CLEVR+ | Geo. |
|---|---|---|---|---|
| base model | 49.1 | 47.7 | 54.9 | 44.8 |
| w/o difficulty control | 60.6 | 48.8 | 81.4 | 51.8 |
| w/ difficulty control | **62.3** | 49.4 | 82.6 | 55.0 |

(d) Data scalability.

| training samples | Avg. | Math | CLEVR+ | Geo. |
|---|---|---|---|---|
| base model | 49.1 | 47.7 | 54.9 | 44.8 |
| 16K | 60.1 | 48.9 | 81.2 | 50.3 |
| 36K | **62.3** | 49.4 | 82.6 | 55.0 |

(e) Input modality.

| input modality | Avg. | Math | CLEVR+ | Geo. |
|---|---|---|---|---|
| base model | 49.1 | 47.7 | 54.9 | 44.8 |
| text | 59.6 | 48.5 | 80.1 | 50.3 |
| vision & text | **62.3** | 49.4 | 82.6 | 55.0 |

(f) SFT vs. RL.

| post-training | Avg. | Math | CLEVR+ | Geo. |
|---|---|---|---|---|
| base model | 49.1 | 47.7 | 54.9 | 44.8 |
| SFT | 47.2 | 38.0 | 71.5 | 32.1 |
| RL | **62.3** | 49.4 | 82.6 | 55.0 |

Table 5: **Ablation study.** We ablate different aspects of ViGaL with Snake and evaluate on downstream benchmarks. The similar evaluation with Rotation is in Sec. B.2. Each benchmark consists of several subtasks (Tab. 2 and Tab. 3), and we report their averages. The base model is Qwen2.5-VL-7B, whose results are in gray. The default settings in Tab. 2 and Tab. 3 are highlighted in blue.

math performance through RL post-training often exhibit substantial drops in general visual capabilities. These results demonstrate that our gameplay-based approach enables math generalization without compromising other visual abilities.

## 3.2 ABLATION STUDY

We ablate key design choices in the Snake environment, evaluate each variant on downstream benchmarks, and report the results in Tab. 5. The corresponding ablation for the Rotation environment is provided in Appendix Sec. B.2.

**Reasoning instructions in the text prompt help.** We use reasoning instructions, such as "`finding the nearest apple by calculating Manhattan distances`", in the text prompts to guide the model thinking chains. The complete text prompts are in Appendix Sec. A.2. In Tab. 5a, we demonstrate that reasoning instructions brings a significant improvement of 2.8%, from 59.5% to 62.3%, for Snake in average accuracy over the three out-of-domain benchmarks.

**Reward design of pre-text game matters for downstream tasks.** We show that reward design of RL for games plays a crucial role for the downstream tasks. As shown in Tab. 5b, we first ask the model to predict only the best next move, defined as the action that moves toward the closest apple while avoiding death. In our improved reward design, we task the model with simultaneously predicting both the best and worst next moves, where the worst move leads directly to losing the game. More importantly, it leads to improvements across all downstream tasks, bringing an average increase of 1.8%. These results suggest that proper reward design in pre-text game can improve not only gameplay capabilities but also generalization to downstream tasks.

Furthermore, inspired by several prior works that improve model performance without labeled rewards (Zhao et al., 2025) or with random labels (Shao et al., 2025), we also provide a random reward ablation, where we still ask the model to predict both best and worst moves but use random moves as the labels. We report the results in the last row in Tab. 5b. In our gameplay setting, RL with random labels reports 49.4% on averagne and does no provide significant gains over the base model, different from the conclusions in prior works (Shao et al., 2025). Potential explanations lie in the difference in data domains and base models, where other works applied random labels to text-only mathematical data while our work applies random labels to visual game data.

**Controlling game difficulty for better reasoning.** Gameplay for RL post-training offers unique opportunities to easily control task difficulty. We present an ablation study on difficulty control importance. We define difficulty based on snake length, where longer snakes represent higher difficulty. For controlled difficulty, we collect training data using states where snake length falls within a moderate range of 1-5. Details are in Sec. A.1. As shown in Tab. 5c, difficulty control achieves 61.4% overall accuracy compared to 60.6% without control. This suggests our game engine can easily generate appropriately difficult data, helping prevent model sub-optimization during RL training.

**RL on games shows data scalability.** Thanks to using game engine, we can generate data at any scale with high flexibility. To show data scalability on RL of visual games, we conduct experiments using 16k and 32k snake game samples, respectively. As in Tab. 5d, scaling data from 16k to

32k brings a performance improvement of 1.3% on average across all domains. This suggests the potential of the proposed ViGaL paradigm to improve downstream performance by easily scaling training data, which contrasts with the data scaling challenges of domain-specific human annotated data, requiring extensive manual effort.

**Both text and vision contribute to better visual reasoning.** To isolate the contributions of text and vision modalities, we conduct an ablation study with a text-only setting. In this setup, we represent game states—including snake positions, apple locations, and boundary constraints—using only textual descriptions during RL training. The model trained with text-only inputs on the Snake game demonstrates substantial improvements across all multimodal benchmarks, with average performance increasing from 49.1% to 59.6%. Incorporating visual inputs yields an additional 1.8% performance gain. These results demonstrate that multimodal RL enhances visual reasoning capabilities, with complementary contributions from both text and vision modalities.

**RL generalizes better than SFT from games to math.** To evaluate the out-of-domain generalization of ViGaL, we compare it with supervised fine-tuning (SFT) using identical visual game data. Tab. 5f shows that SFT with Snake game data degrades the base model's performance on both mathematical reasoning and geometry tasks by a notable 9.7% and 12.7%, respectively. While SFT produces modest improvements on CLEVR+, these gains are substantially smaller than those achieved by RL. Overall, RL improves performance by 12.3%, whereas SFT decreases performance by 1.9%. This stark contrast demonstrates that RL better preserves and extends the model's reasoning capabilities to new domains.

## 4 RELATED WORK

**Reinforcement Learning in MLLMs.** Reinforcement Learning (RL) increasingly enhances reasoning in Large Language Models (LLMs) beyond Supervised Fine-Tuning (SFT). Text-only models like DeepSeek-R1 (DeepSeek-AI, 2025) demonstrate RL's efficacy, especially with rule-based rewards, for complex reasoning. This paradigm is now being extended to Multimodal LLMs (MLLMs). Recent MLLM research explores RL for improved visual reasoning, drawing from LLM successes. Various works (Peng et al., 2025; Huang et al., 2025; Chen et al., 2025b) investigate multi-stage training, trace supervision, or rule-based RL for specific visual subdomains like geometry and counting. Others focus on different RL algorithms like Process Reward Models (PRMs) (Luo et al., 2025; Xiang et al., 2024), often moving beyond SFT-based Chain-of-Thought generation (Dong et al., 2024; Thawakar et al., 2025). Many efforts favor simpler rule-based rewards (Huang et al., 2025; Zhou et al., 2025) over complex reward models prone to hacking (Eisenstein et al., 2023). Unlike approaches training on costly, domain-specific reasoning datasets, our ViGaL paradigm extends rule-based RL to simple, synthetic visual games, demonstrating these serve as scalable, cost-effective pre-text tasks.

**Generalization in MLLMs.** Achieving robust generalization to novel tasks, distributions, and domains is central to MLLM development. RL shows promise for better out-of-distribution (OOD) generalization compared to SFT (Chen et al., 2025b; Meng et al., 2025), and developing multi-step reasoning like CoT (Wei et al., 2022) is itself generalization. This is often pursued through training on large, diverse instruction-following datasets (Li et al., 2024b; Liu et al., 2024; Chen et al., 2024b) or explicitly training general reasoning capabilities (Yang et al., 2025; Huang et al., 2025). While these methods advance OOD generalization, they typically operate within the same broad domain of complex visual reasoning as training data. Our ViGaL paradigm investigates stronger out-of-domain generalization, showing fundamental skills learned from simple synthetic games transfer zero-shot to enhance performance on entirely different, complex domains like visual mathematics and multi-discipline questions, without domain-specific data exposure.

**Transfer Learning and Curriculum Learning.** The idea that training on simpler tasks can improve performance on more complex ones is often framed as transfer learning. In this view, a model first learns from a source task or domain, then reuses that knowledge to improve a different target task (He et al., 2020). Before MLLMs, this principle was studied widely in computer vision through self-supervised pretext tasks, where models are trained without labels to solve auxiliary tasks such

as predicting image rotations (Gidaris et al., 2018), relative positions of patches (Doersch et al., 2015), or colorization, and the learned representations are then transferred to downstream tasks like detection or segmentation. These methods show that solving carefully designed pretext tasks can produce robust visual features that support related vision problems. In reinforcement learning, curriculum learning (Bengio et al., 2009) applies a related idea by ordering tasks or data from easy to hard so that agents gradually master more difficult behaviors. RL curricula and autocurricula have been used to grow complex behaviors from simple environments, for example in multi-agent games and robotics (Baker et al., 2019; Narvekar et al., 2020), typically within a single task family where later tasks share the same goal structure as earlier ones. Our ViGaL paradigm is different in two key respects. First, unlike self-supervised pretext approaches that transfer static feature representations within the visual domain, we transfer learned reasoning policies, such as look-ahead planning, spatial verification, and constraint satisfaction, across tasks and domains. Second, unlike RL curriculum work that mainly increases difficulty within the same environment class, ViGaL transfers policies learned in visual control games to abstract symbolic reasoning tasks in coordinate geometry and multi-discipline question answering, without using domain-specific supervision in those target domains.

## 5 CONCLUSION

We introduced Visual Game Learning (ViGaL), a novel post-training paradigm where MLLMs learn transferable reasoning by playing simple arcade-style games. Our core finding is that RL on games like Snake and Rotation, *without any in-domain math data*, significantly boosts MLLM performance on mathematical and multi-discipline benchmarks, surpassing specialized models and even large proprietary systems. Ablations confirm the importance of game design, reward structure, and that RL outperforms SFT, while distinct games unlock different skills. We posit that games instill fundamental reasoning skills, suggesting a new avenue for using scalable, controllable synthetic games as powerful pre-text tasks to unlock generalizable reasoning. This work opens doors to exploring a broader range of game-based learning for generalizable AI.

## ACKNOWLEDGMENT

We thank Haoqin Tu and Yuxuan Cheng for their valuable feedback on this manuscript.

## ETHICS STATEMENT

This work does not involve human subjects, private or sensitive data, or personally identifiable information. All training and evaluation data are either synthetically generated (Appendix A.1) or come from publicly available benchmarks (Appendix B.6). Our research adheres to the ICLR Code of Ethics, including principles of scientific integrity, fairness, and responsible stewardship. We believe the contributions of this work advance multimodal reasoning without raising foreseeable ethical or societal risks.

**Use of Large Language Models:** An LLM was used to assist with grammar refinement of text. Further details are provided in Appendix D.

## REPRODUCIBILITY STATEMENT

We have taken steps to ensure the reproducibility of our results. The training environments, reinforcement learning setup, and hyperparameters are described in Section 2. Details of synthetic data generation are provided in Appendix A.1, training prompts in Appendix A.2, and reward functions in Appendix A.3. Evaluation protocols for Atari-GPT and visual reasoning benchmarks are specified in Appendix B.1 and Appendix B.6.

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

APPENDIX

**Content**

# A  DATA

## A.1  TRAINING DATA SYNTHESIS

Thanks to using the synthetic game data engine, we can flexibly generate large-scale training data with precisely controlled difficulty levels. This completely eliminates the need for extensive data filtering strategies used in previous rule-based RL work training on domain-specific data like math (Meng et al., 2025; Bae et al., 2025), where difficulty is hard to define and filtering can significantly reduce dataset size.

For the Snake game, the environment consists of a $10 \times 10$ grid game board with two snakes of 1-grid initial length, where at each time step $t$, each snake receives one action to move, resulting in a new game state $s_{t+1}$. We define difficulty based on snake length—longer snakes create more complex game situations and more constrained movement options, closely aligning with how humans perceive difficulty when playing Snake. To generate meaningful moves that accomplish the objective of collecting more apples while remaining alive, we implement a policy network based on Proximal Policy Optimization (PPO) (Schulman et al., 2017). The observation space is represented as a $10 \times 10$ grid with distinct values indicating empty cells (0), apples (1), the agent's own body (2), and other agents' bodies (3), stacked across 4 time steps to incorporate temporal information, resulting in an input tensor $\mathbf{X} \in \mathbb{R}^{10 \times 10 \times 4}$. The policy network architecture consists of two convolutional layers with $3 \times 3$ kernels ($C_1 = 16$ and $C_2 = 32$ output channels), followed by fully connected layers that output action logits for the four possible movements, transformed into a probability distribution $\boldsymbol{\pi}(a|s)$ using softmax. To prevent suicidal moves, we incorporate action priors by masking logits for dangerous actions. The model employs the standard PPO objective with entropy regularization coefficient $\beta = 0.01$, value function coefficient $\lambda = 0.5$, and clipping parameter $\varepsilon = 0.2$. Agents receive rewards of $r = +1$ for collecting apples and penalties of $r = -1$ for dying, enabling them to learn complex behaviors such as obstacle avoidance, apple pursuit, and multi-step trajectory planning.

For the Rotation game, training data comprises synthetically generated visual puzzles focused on 3D spatial reasoning, utilizing 540 unique 3D object meshes (408 from Hunyuan3D 2.0 (Team, 2025) and 132 from Hunyuan3D 2.5). Our custom pipeline produces pairs of images ($I_{\text{init}}, I_{\text{rot}}$) representing objects before and after defined rotations. Difficulty in Rotation is determined by the rotation angle between two images, where smaller angle differences present greater perceptual challenges. Each pair is generated through a precise sequence: establishing diverse initial viewpoints through compound transformations (base orientation plus additional z-axis rotation from $\{0°, 30°, \ldots, 330°\}$ to prevent trivial pattern learning), then applying target rotations of either $90°$ or $180°$ exclusively around the z-axis. All objects are rendered at $512 \times 512$ pixel resolution using a consistent perspective camera under standardized lighting conditions, resulting in approximately 32k unique pairs.

Based on empirical results, we established optimal difficulty parameters for RL training across both games, which we ablate in Tab. 5c. This controlled progression of difficulty, made possible by our synthetic data generation approach, enables more effective learning trajectories compared to traditional data collection methods.

## A.2 TRAINING PROMPT IN VISUAL GAME LEARNING

---

**Prompt for Snake Game**

Your role is to guide a snake within a Snake game featuring multiple apples.

This game is played on a board of size 10 by 10. The board uses a standard Cartesian coordinate system, where (0,0) represents the bottom-left position and (9,9) is the top-rightmost coordinate.

Apples at: {apple_position}

Direction of Your Last Action: {last_action}

Rules:
1) If you move onto an apple, you grow and gain 1 point.
2) If your head moves to a position where its coordinates (x, y) are outside the board boundaries (meaning x < 0, x > 9, y < 0, or y > 9), or into a space occupied by another snake's body, or into a space occupied by your own body, you die. That's the worst move.

**3) The goal is to prioritize snake not die, then efficiently collecting apples. First avoid the worst move, then for each apple, find the nearest apple by calculating Manhattan distances. But only choose best next move to get closer the nearest apple if you can confirm best next move will not run outside the range of the listed coordinates, run into the position of another snake, or yourself. Otherwise it will be the worst move.**

Your snake with the ID {snake_id} in {snake_color} has its head now positioned at {snake_position}, and its body extends to {body_position} You should avoid your next move into your own snake's position.

Enemy snakes in {enemy_color} positions: {enemy_position}.

Decreasing your x coordinate is to the LEFT, increasing your x coordinate is to the RIGHT.
Decreasing your y coordinate is DOWN, increasing your y coordinate is UP.

Read out another snake's position and apple position. Try to predict another snake's next move and avoid colliding with it.

Best answer is one of next move that is the closest to the apple and not lead to your death. Worst answer is all of next moves 1. makes your head's coordinates (x, y) are outside the board boundaries, meaning x < 0, x > 9, y < 0, or y > 9. 2. moves into a position occupied by another snake's body. 3. moves into a position occupied by body of yourself.

Check all the next moves to list out all the worst moves in <worst_answer> tag. If no worst answer, return None for worst answer, e.g., "<worst_answer>None</worst_answer>"

The best answer and the worst answer are mutually exclusive and different.

You need first to give your reasoning process then to choose one of best next move and worst next move from ['UP', 'DOWN', 'LEFT', 'RIGHT'].

The reasoning process and answer are enclosed within <think> </think>, <best_answer> </best_answer> and <worst_answer> </worst_answer> tags, respectively, i.e., "<think> reasoning process here </think><best_answer> one best move here </best_answer><worst_answer> all worst moves here </worst_answer>"

---

---

**Prompt for Rotation Game**

I'm showing you 4 images. Images 1-2 are an example pair, and Images 3-4 are the test pair. In each pair, the first image shows the initial orientation, and the second shows the object after rotation.

### EXAMPLE OF ROTATION ###

Example: Image 1 shows the initial view and Image 2 shows the object after a 180 degree rotation.

### YOUR TASK ###
Now, considering the transformation from Image 3 (initial) to Image 4 (rotated)
. Determine the angle of rotation from Image 3 to Image 4 on the plane
Analyze the rotation carefully using the example pair (Images 1-2) as a reference.

**1. Coordinate System Transformation:**
**- Draw an x-y coordinate system on both original and rotated images with origin at center**
**- Identify a distinct feature point and note its coordinates in both images**
**- Apply rotation matrix equations to verify the transformation**

 **Example: A star icon at coordinates (3,1) in the original image appears at (-1,3) in the rotated image. Testing with the 90° clockwise rotation matrix [cos(90°), sin(90°); -sin(90°), cos(90°)] confirms the transformation from (3,1) to (-1,3), verifying a 90° clockwise rotation.**

**2. Angular Displacement Measurement:**
**- Mark the image center as the origin in both images**
**- Draw a straight line from center to a distinctive feature in both images**
**- Measure the angle between these two lines using counterclockwise as positive**

 **Example: A line from center to a red dot makes a 30° angle with horizontal in the original image. In the rotated image, this line makes a 210° angle with horizontal. The difference (180°) indicates a clockwise 180° rotation.**

**3. Symmetry Axis Tracking:**
**- Identify major symmetry axes in the original image**
**- Locate the same symmetry axes in the rotated image**
**- Calculate the angular displacement between original and rotated axes**

 **Example: A rectangular logo has vertical and horizontal symmetry axes. After rotation, the vertical axis now points right and horizontal points down. This 90° shift of both axes confirms a clockwise 90° rotation.**

**4. Triangle Configuration Analysis:**
**- Select three non-collinear distinct points forming a triangle in both images**
**- Compare the orientation of this triangle in both images using vector cross products**
**- Determine rotation angle from the triangle's orientation change**

 **Example: Three points form a right triangle with vertices clockwise arranged. After rotation, the same triangle has its vertices arranged in counterclockwise order while maintaining the same shape. This inversion indicates a clockwise 180° rotation.**

**5. Polar Coordinate Comparison:**
**- Convert key points to polar coordinates (r,$\theta$) relative to image center**
**- Compare $\theta$ values of the same features in original and rotated images**
**- Calculate consistent angular difference across multiple points**

 **Example: A feature at polar angle 45° in the original image appears at 135° in the rotated image. Another feature shifts from 10° to 100°. Both show a +90° shift in polar angle, confirming a clockwise 90° rotation.**

Choose the rotation angle from this list: ['counter clockwise 90', '180']

The reasoning process and answer are enclosed within <think> </think> and <answer> </answer> tags, respectively, i.e., "<think> reasoning process here </think><answer> answer here </answer>"

While the model takes images as input to understand the current state of the game, we design a structural text prompt framework to also provide game guidance. Our game prompts consist of two parts: (1) game settings and (2) reasoning instructions. (1) To help the model understand the game environment, we describe the background, current game state, rules, goals, action space, *etc.* in text besides the input image. (2) In the reasoning instruction part, we provide specific thinking guidance since games can be approached with various thinking chains. To encourage broader thinking, we implement different types of reasoning instructions to guide decision-making process. Specifically, we used GPT-4o (Hurst et al., 2024) to synthesize mathematical thinking instructions for Snake, such as "`finding the nearest apple by calculating Manhattan distances`", and spatial thinking instructions for Rotation, for example, "`identify major symmetry axes in the original image`". With reasoning instructions for games, the obtained reasoning abilities generalize to downstream evaluation on visual math questions (Tab. 5a). **Bold text** indicates reasoning instructions synthesized by GPT-4o (Hurst et al., 2024).

### A.3 DETAIL OF FORMAT REWARD

The format reward $r_{\text{format}}$ validates whether the response follows the task-specific format:

$$r_{\text{format}} = \begin{cases} 0.1, & \text{if the response follows the required format} \\ 0, & \text{otherwise} \end{cases} \tag{1}$$

For Snake game, the desired format is:

`<think>...</think><best_answer>...</best_answer><worst_answer>...</worst_answer>`.

As suggested by the format, we encourage the model to predict both a positive move that moves toward the apple and a negative move that leads to failure. This reward encourages contrastive decision-making, which not only improves the model's gameplay abilities but also boosts downstream reasoning performance on visual math benchmarks. We ablate the effect in Tab. 5b. For the rotation task, the required format is simply `<think>...</think><answer>...</answer>`.

# B EVALUATION

## B.1 EVALUATION DETAIL OF ATARI GAME

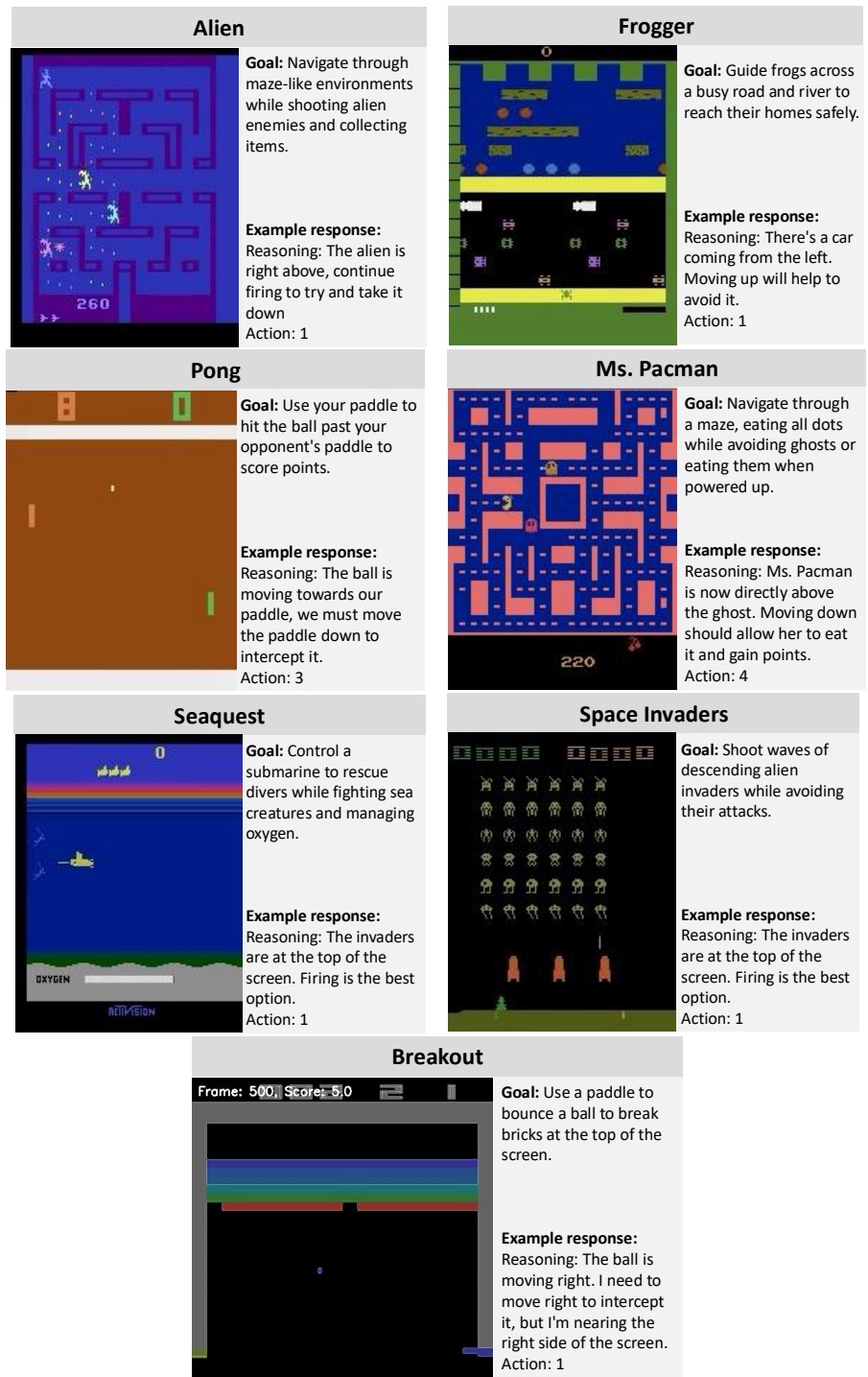

Figure 5: **Goal and example response from model of Atari games used for evaluation.** We implement 7 kinds of Atari games from Atari-GPT (Waytowich et al., 2024).

To evaluate out-of-distribution generalization, we test ViGaL on Atari-GPT (Waytowich et al., 2024), a benchmark for evaluating MLLMs as decision-making agents in Atari video games, as shown in Fig. 5. The benchmark consists of seven different Atari games: Alien, Frogger, Pong, Ms. Pacman, Seaquest, Space Invaders and Breakout. These games present diverse visual environments which is different from Snake game and Rotation game, and require different strategic approaches to finish the goal, making them an ideal test bed for ViGaL evaluating out-of-distribution generalization capabilities.

For evaluation, we input game frames as pixel observations to our model, following the established protocol in Atari-GPT. Specifically, each game frame is resized from $210 \times 160 \times 3$ to $512 \times 512 \times 3$, then provided to our model along with game-specific action information. We maintain a context buffer containing the two previous frames and responses together with the current frame to enable temporal reasoning. Following Atari-GPT, we implement frame skipping of 8 frames, which extends the standard 4-frame skipping in ALE to reduce computational intensity while preserving gameplay continuity.

We evaluate our method through four independent rollouts of 1,000 timesteps each and report the average cumulative reward, with results presented in Tab. 1c.

## B.2 Ablation On Rotation Game

Table 6: **Ablation study.** Similar to the evaluation in Tab. 5, we analyze how different aspects of our post-training strategy within the Rotation game affect downstream generalization benchmarks. The base model is Qwen2.5-VL-7B, with results shown in gray. The default settings from Tab. 2 and Tab. 3 are highlighted in blue. We observe the same improvement trends for each strategy as reported in Tab. 5.

<table>
<tr><td colspan="5" align="center">(a) Prompt design.</td><td colspan="5" align="center">(b) SFT vs. RL.</td></tr>
<tr><th>prompt</th><th>Avg.</th><th>Math</th><th>CLEVR+</th><th>Geo.</th><th>post-training</th><th>Avg.</th><th>Math</th><th>CLEVR+</th><th>Geo.</th></tr>
<tr><td>base model</td><td>49.1</td><td>47.7</td><td>54.9</td><td>44.8</td><td>base model</td><td>49.1</td><td>47.7</td><td>54.9</td><td>44.8</td></tr>
<tr><td>w/o Reasoning Instruction</td><td>61.4</td><td>48.9</td><td>80.4</td><td>54.8</td><td>SFT</td><td>55.6</td><td>44.0</td><td>75.4</td><td>47.5</td></tr>
<tr><td>w/ Reasoning Instruction</td><td>62.6</td><td>49.3</td><td>80.7</td><td>57.9</td><td>RL</td><td>62.6</td><td>49.3</td><td>80.7</td><td>57.9</td></tr>
</table>

<table>
<tr><td colspan="5" align="center">(c) Difficulty control.</td></tr>
<tr><th>difficulty control</th><th>Avg.</th><th>Math</th><th>CLEVR+</th><th>Geo.</th></tr>
<tr><td>base model</td><td>49.1</td><td>47.7</td><td>54.9</td><td>44.8</td></tr>
<tr><td>w/o difficulty control</td><td>61.0</td><td>48.0</td><td>80.2</td><td>54.8</td></tr>
<tr><td>w/ difficulty control</td><td>62.6</td><td>49.3</td><td>80.7</td><td>57.9</td></tr>
</table>

As shown in Tab. 6, we conduct a similar ablation study to Tab. 5, but replace the Snake game environment with the Rotation game. Our results demonstrate the same consistent improvement trends on downstream generalization benchmarks for each strategy employed.

Specifically, we control the task difficulty by varying the rotation angles between two images. In the uncontrolled difficulty setting, the rotation angle between images can be clockwise 90°, counter-clockwise 90°, or 180°. However, we found that explicitly requiring the model to distinguish between clockwise and counter-clockwise rotations leads to training difficulties. Therefore, we remove it and only retain option of clockwise 90° and 180° rotations.

Unlike the Snake game, we cannot conduct the ablations shown in Tab. 5e because the Rotation game is inherently vision-dependent and requires visual input. Similarly, we cannot perform the ablations in Tab. 5b because the Rotation game provides only binary answer options, making it impossible to meaningfully designate both "best" and "worst" answers simultaneously.

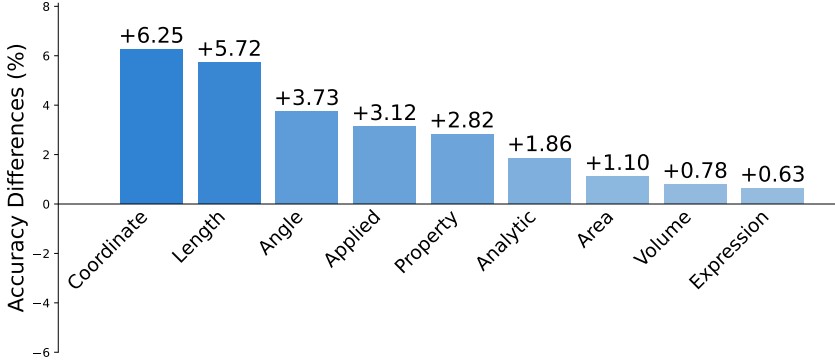

Figure 6: **Accuracy differences between ViGaL-Snake+Rotation and base model without RL training across mathematical subfields in Mathverse.** The synergistic effects of jointly training on two games observed suggest that complementary games can enhance overall mathematical reasoning capabilities.

### B.3 Synergistic Effects of Multi-Game Training

As discussed in Sec. 3.1, our analysis reveals that each game develops distinct reasoning abilities in the model. To investigate potential combined benefits, we conducted experiments where models were trained simultaneously on *both* the Snake and Rotation games. Fig. 6 shows that joint training effectively combines the strengths of each individual game, improving performance across the mathematical areas where each game shows particular effectiveness, resulting in greater overall gains on Mathverse. These results suggest that strategically combining games with complementary strengths offers a simple yet effective approach to enhance model generalization abilities.

### B.4 Generalization Beyond Snake and Rotation: A Multi-Game Study

A key question is whether the observed benefits are specific to Snake and Rotation, or whether they generalize across diverse game mechanics. To investigate this, we extend our study to four additional games: Sokoban, Maze, Tetris, and Sudoku (Tong et al., 2025). Sokoban requires pushing boxes to target positions while navigating spatial constraints; Maze involves pathfinding through obstacles to reach a goal; Tetris requires rotating and positioning falling shapes to complete rows; Sudoku demands filling grids while satisfying row, column, and box constraints. These games introduce cognitive challenges beyond Snake and Rotation, including strategic planning and logical deduction. This extended study allows us to examine whether the transfer of reasoning skills from gameplay to mathematical tasks is a general phenomenon or an artifact of our initial game selection.

**Consistent improvements across diverse games.** Tab. 7 shows that all games yield consistent gains across mathematical reasoning benchmarks, with improvements ranging from 1.5% to 1.7% on average. This demonstrates that developing transferable reasoning skills from visual gameplay is robust across diverse game mechanics and not specific to Snake and Rotation.

**Systematic transfer patterns through subject-level analysis.** To understand how different game mechanics transfer to specific mathematical sub-skills, we analyze performance on MathVerse using its official subject split, which assigns each problem to one of three subjects: Plane Geometry (angles and lengths), Functions (coordinate graphs and related diagrams), and Solid Geometry (volume and surface area). We use this subject-level split instead of the finer subfields because many of those subfields depend on overlapping skills. For example, the Coordinate, Property, and Expression subfields all require interpreting coordinate systems. The subject-level grouping therefore offers a more stable and disentangled proxy for distinct mathematical reasoning abilities, for instance by placing all questions that require interpreting coordinate systems under Functions.

Tab. 8 reports performance gains for all six games separately for each subject. Based on these performance profiles, we grouped the six games into three clusters using K-Means clustering to identify systematic transfer patterns:

Table 7: Comparison of different auxiliary games on mathematical reasoning benchmarks. All models use Qwen2.5-VL-7B as the base. All games yield consistent improvements, demonstrating the generalizability of game-based transfer learning.

| Model | Math Avg. | MathVerse | MathVista | MathVision |
|---|---|---|---|---|
| Qwen2.5-VL-7B | 47.7 | 49.0 | 68.0 | 26.0 |
| + Maze | 49.3 | 50.9 | 70.3 | 26.7 |
| + Rotation | 49.3 | 50.4 | 71.2 | 26.3 |
| + Snake | 49.4 | 51.1 | 70.7 | 26.5 |
| + Sokoban | 49.3 | 50.5 | 70.1 | 27.2 |
| + Sudoku | 49.3 | 51.2 | 69.0 | 27.6 |
| + Tetris | 49.2 | 50.8 | 69.4 | 27.4 |

Table 8: Performance gains on MathVerse subjects for each auxiliary game, showing accuracy improvement relative to the Qwen2.5-VL-7B baseline. We use the official subject split from Math-Verse: Plane Geometry (angles and lengths), Functions (coordinate graphs and related diagrams), and Solid Geometry (volume and surface area). This subject-level grouping provides a more stable proxy for distinct mathematical reasoning abilities compared to finer subfields that depend on overlapping skills.

| Model | Plane Geometry | Functions | Solid Geometry |
|---|---|---|---|
| Baseline (Qwen2.5-VL-7B) | 52.43 | 49.81 | 33.45 |
| +Snake | +1.81 | +3.77 | +0.84 |
| +Rotation | +2.32 | +1.13 | +0.50 |
| +Maze | +1.96 | +1.76 | +1.84 |
| +Tetris | +1.73 | +2.01 | +1.34 |
| +Sokoban | +1.49 | +1.89 | +0.84 |
| +Sudoku | +2.28 | +2.27 | +1.34 |

- **Cluster 1 (Snake, Sokoban)**: Strongest transfer to **Functions** (+2.83% avg). Snake and Sokoban require tracking multiple objects moving in coordinate space: the snake's growing body segments or boxes being pushed to target positions. Players must reason about how coordinate relationships between objects evolve as actions unfold, mirroring the core skill in Functions problems, understanding coordinate transformations (e.g., translations, scaling) and how values change under sequential operations.

- **Cluster 2 (Rotation)**: Strongest transfer to **Plane Geometry** (+2.32%). This puzzle requires recognizing 3D object rotation angles, directly engaging angle reasoning and spatial relationships fundamental to plane geometry.

- **Cluster 3 (Maze, Sudoku, Tetris)**: Balanced improvements across all three subjects. These games engage distinct reasoning modes: Maze requires navigating to find optimal paths; Sudoku demands symbolic constraint satisfaction without spatial movement; Tetris involves shape transformations and pattern completion. This diversity of non-overlapping cognitive demands supports reasoning across mathematical domains more broadly rather than specializing to a single subject.

These findings reveal systematic relationships between game mechanics and mathematical reasoning categories. Beyond confirming the generalizability of our approach, the results provide practical guidance for auxiliary task selection: to improve a specific downstream skill, one can select existing games whose cognitive requirements align with the target skill. Our framework offers a method for

systematically choosing and combining off-the-shelf games based on observed cognitive transfer patterns.

## B.5 REASONING ABILITY BOUNDARY VIA PASS@$k$ EVALUATION

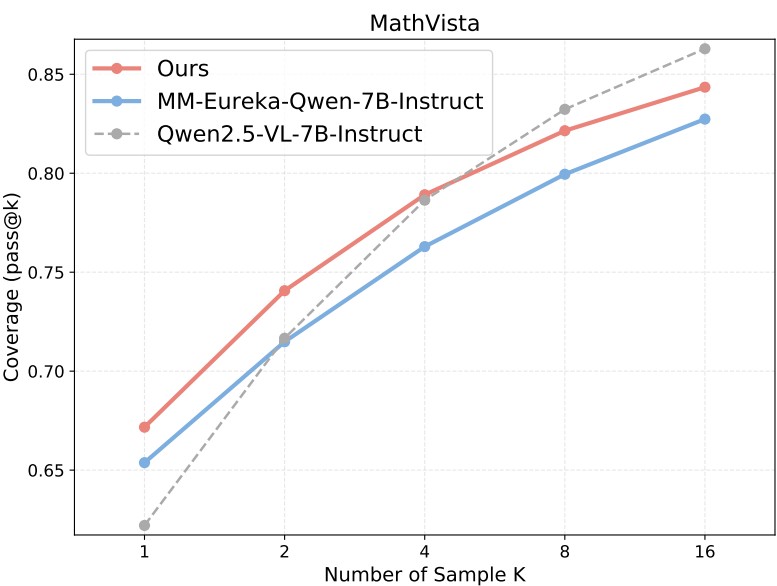

Figure 7: Pass@$k$ performance curves on MathVista comparing base models with their zero-RL counterparts trained on mathematical data and game data, respectively.

We explore the reasoning ability boundary of models trained with different RL approaches by evaluating the pass@$k$ metric. This metric measures the probability that at least one of $k$ independent model samples solves a given problem, indicating the true scope or boundary of a model's reasoning capability - essentially what problems the model can potentially solve given enough sampling attempts.

We evaluate the pass@$k$ performance of three models: the Base Model without RL training, MM-Eureka-Qwen-7B-Instruct, and our ViGaL. As shown in Fig. 7, our ViGaL consistently demonstrates increasing pass@$k$ scores on Mathverse as $k$ increases. This finding suggests that our approach can effectively solve complex problems when allowed multiple reasoning attempts, uncovering capabilities not apparent in single-sample evaluations.

Moreover, compared to the other RL-trained model, MM-Eureka-Qwen-7B-Instruct, our model achieves a steeper improvement in pass@$k$ as $k$ increases. This indicates that ViGaL possesses a broader reasoning boundary and stronger reasoning abilities, enabling it to solve a wider range of problems when given sufficient opportunities to explore different solution paths.

Finally, our results demonstrate that as $k$ increases, base models without RL training eventually outperform RL-trained models. This aligns with the findings in (Yue et al., 2025) that highlight a fundamental limitation of reinforcement learning with verifiable rewards (RLVR): while RL training significantly improves performance at small $k$ values (e.g., pass@1), base models possess a wider coverage of solvable problems. This suggests a trade-off where RL optimization focuses on solving high-probability problems at the expense of broader solution coverage. Future work should explore RLVR algorithms that can improve pass@$k$ performance across all values of $k$, effectively extending the reasoning boundary beyond that of the base model.

## B.6 DETAIL OF EVALUATION BENCHMARKS

To obtain a clearer picture of the various facets of MLLM performance, we follow prior studies (Tong et al., 2024a; Li et al., 2024c) and systematically and carefully divide existing benchmarks

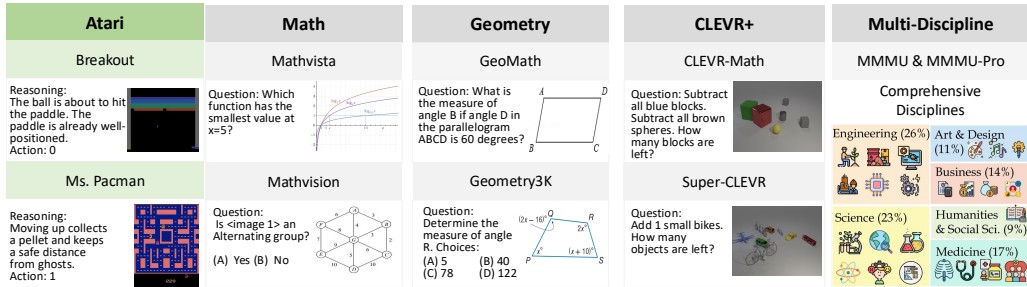

(a) Atari games.  (b) Out-of-domain tasks.

Figure 8: **Samples from our generalization reasoning benchmarks.** We evaluate the proposed Vi-GaL with two types of generalization: (a) *out-of-distribution* generalization, where models trained on our visual games are tested on unseen Atari games (Waytowich et al., 2024); and (b) *out-of-domain* generalization, where models trained only on game tasks are evaluated on diverse multi-modal reasoning tasks including mathematical reasoning, geometric problem-solving, 3D under-standing on CLEVR+ and multi-discipline reasoning on MMMU series.

into two broad groups: (i) *reasoning-oriented benchmarks*, which require multi-step or mathematical reasoning to solve the problems, and (ii) *general-purpose perception benchmarks*, which primarily assess broad visual understanding and perception abilities.

For reasoning-oriented benchmarks, we comprehensively evaluate the visual reasoning generalization capabilities of RL through gaming on a diverse collection of tasks that specifically demand advanced visual reasoning skills, including math-focused tasks like Math and Geometry, and other comprehensive reasoning benchmarks beyond math, like CLEVR+ and Multi-Discipline. Fig. 8b illustrates specific examples from each benchmark.

- **Math** evaluates multimodal math reasoning with widely-used datasets: MathVista (test-mini) (Lu et al., 2024), MathVerse (testmini) (Zhang et al., 2024), and MathVision (test) (Wang et al., 2024b). MathVista offers diverse problems spanning VQA, logic, algebra, and geometry; MathVerse emphasizes algebraic and geometric image comprehension; MathVision tests abstract visual reasoning.

- **Geometry** evaluates structural interpretation skills across mathematical diagrams, medical images, charts, and architectural layouts. It uses datasets GeoMath (Geo170K (Gao et al., 2023), Math360K (Shi et al., 2024)) and Geometry3K (Lu et al., 2021), featuring both choice and non-choice questions. Following Reason-RFT (Tan et al., 2025), we test with 820 GeoMath and 800 Geometry3K samples.

- **CLEVR+** evaluates the integration of mathematical and spatial reasoning skills through challenging arithmetic problems in complex 3D block-based scenes, including sub-tasks on CLEVR-Math (Lindström & Abraham, 2022) and Super-CLEVR (Li et al., 2023). Following Reason-RFT (Tan et al., 2025), we use 1K test samples from each of CLEVR-Math and Super-CLEVR.

- **Multi-Discipline** evaluates college-level expert knowledge across six disciplines: Art & Design, Business, Science, Health & Medicine, Humanities & Social Science, and Tech & Engineering. We follow the evaluation setting of MMMU (Yue et al., 2024a) val set (900 questions) and MMMU-Pro (Yue et al., 2024b) overall score (average of standard 10-option and vision-only settings).

For general-purpose perception benchmarks, we systematically evaluate comprehensive visual capabilities. Following previous work, these benchmarks are categorized into three distinct types: General, Vision-Centric, and OCR & Chart.

- **General** benchmarks assess fundamental visual understanding capabilities. We evaluate MuirBench (Wang et al., 2024a) for multi-image understanding and CRPE (Kazemzadeh et al., 2014) for relation understanding.

| Model | Avg. | General | | | Vision-Centric | | | | | | OCR & Chart | | | | |
|---|---|---|---|---|---|---|---|---|---|---|---|---|---|---|---|
| | | Avg. | Muir-Bench | CRPE_rel. | Avg. | MMVP | Real-WorldQA | MMStar | BLINK_val | MME_p | Avg. | AI2D w.M. | SEED-Bench-2+ | DocVQA val | OCR-Bench |
| Proprietary Model | | | | | | | | | | | | | | | |
| GPT-4o (Hurst et al., 2024) | 74.8 | 72.3 | 68.0 | 76.6 | 69.4 | – | 75.4 | 64.7 | 68.0 | 1614 | 82.6 | 84.6 | 72.0 | 91.1 | 736 |
| General Multimodal Language Model | | | | | | | | | | | | | | | |
| Qwen2.5-VL-7B (Bai et al., 2023) | 72.4 | 68.0 | 59.6 | 76.4 | 65.8 | 74.3 | 68.5 | 63.9 | 56.4 | 1698 | 83.3 | 83.9 | 70.4 | 95.7 | 864 |
| Multimodal Reasoning Model Post-Trained on Qwen2.5-VL-7B | | | | | | | | | | | | | | | |
| R1-Onevision-7B (Yang et al., 2025) | – | 66.8 | 46.3 | 87.3 | 56.5 | 61.3 | 58.0 | 57.8 | 48.7 | 1504 | – | – | – | – | – |
| R1-VL-7B (Chen et al., 2025b) | 67.4 | 63.3 | 54.1 | 72.4 | 59.6 | 70.3 | 61.4 | 55.6 | 51.0 | 1657 | 79.2 | 81.7 | 66.4 | 89.4 | 81.0 |
| MM-Eureka-Qwen-7B (Meng et al., 2025) | 71.8 | **68.9** | 61.1 | 76.7 | 65.1 | 74.3 | 66.1 | 65.9 | 54.0 | 1626 | 81.5 | 84.3 | 68.2 | 92.0 | 87.0 |
| Reason-RFT-Zero-7B (Tan et al., 2025) | 68.4 | 66.9 | 58.5 | 75.2 | 58.5 | 58.0 | 65.3 | 59.1 | 51.6 | 1653 | 79.8 | 83.3 | 68.0 | 88.1 | 82.0 |
| VLAA-Thinker-7B (Chen et al., 2025a) | 69.7 | 65.9 | 57.1 | 74.6 | 62.6 | 71.6 | 65.4 | 60.4 | 53.0 | 1593 | 80.6 | 83.4 | 67.4 | 90.9 | 84.5 |
| OpenVLThinker-7B (Deng et al., 2025) | – | 64.3 | 52.8 | 75.8 | 50.4 | 32.3 | 60.2 | 59.1 | 49.9 | 1513 | – | – | – | – | – |
| ViGaL Snake + Rotation | **72.2** | 68.6 | 60.5 | 76.7 | **65.7** | 74.6 | 67.3 | 65.4 | 55.6 | 1685 | **82.2** | 84.8 | 69.1 | 92.7 | 86.6 |

Table 9: **Main results on multimodal language benchmarks targeting more general and comprehensive visual ability.** We compare with models post-trained on Qwen2.5-VL-7B (Bai et al., 2023). Best category averages are highlighted in **bold**. Note that $MME_p$ is excluded from vision-centric category average accuracy due to scale differences.

- **Vision-Centric** benchmarks thoroughly evaluate perception, real-world understanding, and multi-modal capabilities. We assess MMVP (Tong et al., 2024b), RealWorldQA (X.AI, 2024), MMStar (Chen et al., 2024a), MME (Fu et al., 2023), and BLINK (Fu et al., 2024).

- **OCR & Chart** understanding benchmarks focus on text-rich visual content. We specifically use AI2D (Kembhavi et al., 2016) for diagram understanding, SEED-Bench-2-Plus (Li et al., 2024a) for text-rich visual comprehension, DocVQA (Mathew et al., 2021) for document understanding, and OCRBench (Liu et al., 2023) for comprehensive OCR evaluation.

## B.7 EVALUATION ON GENERAL VISUAL CAPABILITIES

As discussed in Sec. 3.1, we have already demonstrated that our game RL training can generalize to visual math reasoning and multi-discipline reasoning benchmarks. To evaluate whether reasoning improvements come at the cost of general visual understanding, we test ViGaL Snake + Rotation on diverse general visual benchmarks. The results, summarized in Tab. 9, show that our model preserves general visual performance at a level comparable to the Qwen2.5-VL-7B base model, while delivering stronger gains in mathematical reasoning. In contrast, prior RL-based approaches often sacrifice visual accuracy when optimized for reasoning. This confirms that our gameplay-driven strategy enhances reasoning ability without compromising broader visual competencies.

## B.8 INFERENCE LENGTH ANALYSIS

Recent reinforcement learning studies (Xie et al., 2025; Aggarwal & Welleck, 2025) have raised questions about whether performance improvements stem from genuinely enhanced reasoning capabilities or merely from models generating longer responses. To address this concern, we analyze the relationship between response length and performance for models trained with our game-based approach.

| Model | Response Length | Math Avg. |
|---|---|---|
| Qwen2.5-VL-7B (baseline) | 250 | 47.7 |
| ViGaL (ours, RL on games) | 268 | **50.6** |

Table 10: Response length and performance on visual math benchmarks. Our game-based RL approach achieves significant performance gains while maintaining comparable inference costs.

Table 10 demonstrates that our performance improvements are not simply due to increased verbosity. Our ViGaL model achieves substantial performance gains (50.6% vs. 47.7%) while maintaining nearly identical inference costs—the response length increases by only 7% (268 vs. 250 tokens). This minimal increase in response length, coupled with the significant accuracy improvement, indicates that the model has learned transferable skills rather than merely generating longer outputs.

These results suggest that game-based RL training enables effective knowledge transfer from game environments to mathematical problem-solving. For example, spatial reasoning skills acquired from the Rotation game and coordinate recognition abilities developed through Snake gameplay transfer effectively to visual math tasks. The model thus learns genuine problem-solving strategies while maintaining inference costs.

### B.9    Reasoning Correlation Analysis Between Game and Math

To understand the mathematical reasoning patterns in snake game playing, we developed a systematic approach to extract and analyze reasoning steps from multiple gameplay traces. Our methodology uses GPT-5 OpenAI (2025) as an analytical tool in a two-stage process.

In Stage A, we collect multiple snake game "thinking traces", which are detailed reasoning sequences generated during gameplay, and distill them into a generalized set of 8 core reasoning steps. These steps abstract away specific details like exact coordinates or particular board configurations to capture fundamental cognitive operations. The operations include parsing board state, enumerating moves, safety screening, path metric selection, distance computation, target identification, enemy anticipation, and move ranking. This summarization ensures our analysis focuses on transferable reasoning patterns rather than game-specific instances.

In Stage B, we quantify how mathematical each reasoning step is by evaluating its correlation with nine distinct mathematical aspects. We use a simple 3-level scoring system where 0 means no correlation, 1 means low correlation, and 2 means high correlation. GPT-5 analyzes how strongly each step relates to mathematical concepts such as coordinate manipulation, distance metrics, analytical reasoning, and geometric properties.

The resulting correlation matrix in Tab. 11 reveals clear patterns. Coordinate-based reasoning dominates steps that involve spatial parsing and movement planning, particularly Steps 1 through 3, Step 5, and Step 7. Meanwhile, analytical and length-based reasoning become prominent in optimization steps like target identification and move ranking, seen in Steps 6 and 8. Steps 4 and 5, which involve path metrics and distance computation, show high correlation with both coordinate systems and length calculations. This confirms the geometric nature of pathfinding in grid-based environments. Our systematic analysis demonstrates that even seemingly simple game-playing behaviors require sophisticated integration of multiple mathematical reasoning capabilities.

---

**Prompt Template for Reasoning Step Extraction and Correlation Analysis**

**Stage A - Step Extraction:**
Given multiple snake game thinking traces, extract N general reasoning steps (6-9 steps) that capture the core operations. Abstract away instance-specific details and output:

- Short, action-oriented step names with one-line descriptions
- General patterns covering: state parsing, move generation, safety screening, target selection via distance, opponent awareness, scoring/tie-breaks, decision, reporting

**Stage B - Mathematical Aspect Correlation:**
For each extracted step, assign correlation levels (0/1/2) to these mathematical aspects:

- **Expression**: Formatting/structuring outputs
- **Coordinate**: Reading/writing positions, mapping moves to (x,y)
- **Area**: Board regions/bounds as areas
- **Volume**: 3D spatial reasoning (if applicable)
- **Applied**: Goal-directed task execution
- **Property**: Rules/invariants (bounds, occupancy, collision)
- **Angle**: Angle-based path reasoning
- **Analytic**: Selection/optimization, tie-break logic
- **Length**: Distance metrics (Manhattan/L1, grid paths)

Output as structured table with integer scores only (0 = no correlation, 1 = low, 2 = high).

---

.

Table 11: Correlation Matrix of each step reasoning trace of playing snake game with solving math questions. (0=No Correlation, 1=Low Correlation, 2=High Correlation)

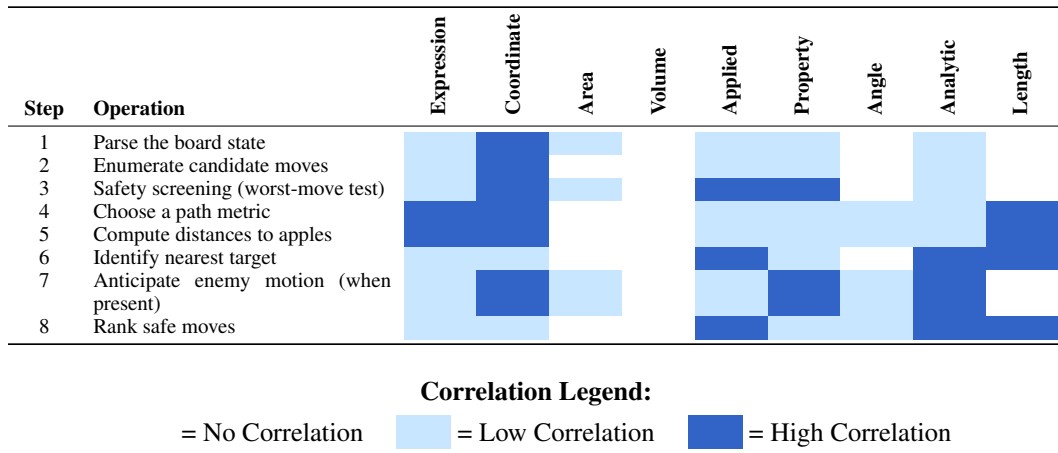

| Step | Operation |
|---|---|
| 1 | Parse the board state |
| 2 | Enumerate candidate moves |
| 3 | Safety screening (worst-move test) |
| 4 | Choose a path metric |
| 5 | Compute distances to apples |
| 6 | Identify nearest target |
| 7 | Anticipate enemy motion (when present) |
| 8 | Rank safe moves |

**Correlation Legend:**

= No Correlation    = Low Correlation    = High Correlation

### B.10 GENERALIZATION TO 3D SPATIAL REASONING (VSI-BENCH)

| Model | Avg. | Obj. Count | Abs. Dist. | Obj. Size | Room Size | Rel. Dist. | Rel. Dir. | Route Plan | Appr. Order |
|---|---|---|---|---|---|---|---|---|---|
| *Proprietary Model* | | | | | | | | | |
| GPT-4o (Hurst et al., 2024) | 34.0 | 46.2 | 5.3 | 43.8 | 38.2 | 37.0 | 41.3 | 31.5 | 28.5 |
| Gemini-1.5-Flash (**?**) | 42.1 | 49.8 | 30.8 | 53.5 | 54.4 | 37.7 | 41.0 | 31.5 | 37.8 |
| Gemini-1.5-Pro (**?**) | 45.4 | 56.2 | 30.9 | 64.1 | 43.6 | 51.3 | 46.3 | 36.0 | 34.6 |
| *Open-source Models* | | | | | | | | | |
| InternVL2-8B (Chen et al., 2024b) | 34.6 | 23.1 | 28.7 | 48.2 | 39.8 | 36.7 | 30.7 | 29.9 | 39.6 |
| InternVL2-40B (Chen et al., 2024b) | 36.0 | 34.9 | 26.9 | 46.5 | 31.8 | 42.1 | 32.2 | 34.0 | 39.6 |
| LongVILA-8B (**?**) | 21.6 | 29.1 | 9.1 | 16.7 | 0.0 | 29.6 | 30.7 | 32.5 | 25.5 |
| VILA-1.5-40B (**?**) | 31.2 | 22.4 | 24.8 | 48.7 | 22.7 | 40.5 | 25.7 | 31.5 | 32.9 |
| LongVA-7B (**?**) | 29.2 | 38.0 | 16.6 | 38.9 | 22.2 | 33.1 | 43.3 | 25.4 | 15.7 |
| *Multimodal Reasoning Model Post-Trained on Qwen2.5-VL-7B (**?**)* | | | | | | | | | |
| *Base Model (Qwen2.5-VL-7B)* | 36.7 | 41.9 | 21.4 | 50.4 | 36.8 | 38.5 | 40.9 | 29.9 | 34.1 |
| Video-R1 (**?**) | 35.1 | – | – | – | – | – | – | – | – |
| Visual Jigsaw (**?**) | 38.5 | – | – | – | – | – | – | – | – |
| ViGaL | **38.7** | 42.9 | 22.6 | 46.7 | 45.8 | 38.3 | 41.7 | 35.1 | 36.1 |

Table 12: **Results on VSI-Bench 3D spatial reasoning.** We compare proprietary models, open-source multimodal models, and multimodal reasoning models post-trained on Qwen2.5-VL-7B.

Tab. 12 demonstrates that post-training on two games using synthesized 2D images successfully generalizes to 3D spatial reasoning tasks. On VSI-Bench (**?**), ViGaL improves by 2.0% after game-based post-training, slightly surpassing Visual Jigsaw (**?**). This is notable because Visual Jigsaw is post-trained on 3D video data, whereas our approach uses only 2D game data. These results further support the hypothesis that gameplay-based RL post-training develops transferable spatial reasoning skills that extend beyond 2D visual domains.

# C    CASE STUDY

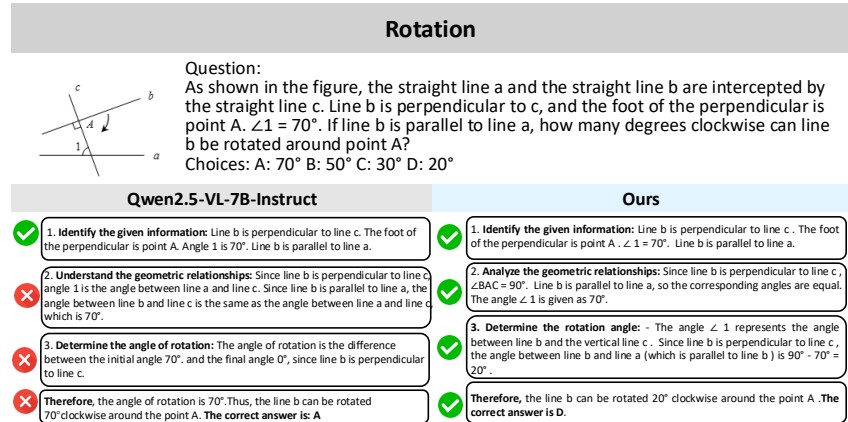

(a) A case study from Mathverse. Base model misinterpreted the geometric configuration and rotation direction, while our model correctly identified the perpendicular relationship and calculated the proper angle.

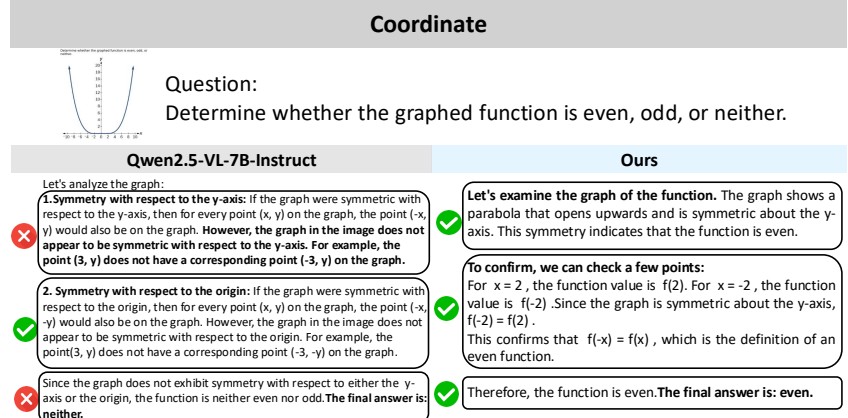

(b) A case study from Mathverse. Base model misperceived critical visual information like symmetry and coordinates in graphs, while our model demonstrated accurate visual perception for mathematical elements.

Figure 9: Comparison of base model and our model after rule-based RL training, showing improved visual-mathematical reasoning on geometric and coordinate problems.

We provide quantitative comparison examples below to demonstrate reasoning improvements on mathematical problems after RL training. In Fig. 9a, when solving a geometric angle problem, the base model fails to correctly interpret the critical relationship between perpendicular lines and corresponding angles. It makes contradictory assumptions about angle measures, leading to an incorrect calculation of the required rotation. In contrast, our ViGaL precisely tracks the geometric constraints and properly calculates the angle difference between initial and target positions. In Fig. 9b, when analyzing function properties from a graph, the base model incorrectly claims the function lacks symmetry despite clear visual evidence. It fails to recognize the fundamental y-axis symmetry of the parabola shown in the image. Our model immediately identifies this critical symmetrical pattern and correctly applies the appropriate mathematical definition of an even function, demonstrating enhanced visual perception of mathematical structures.

# D    USE OF LARGE LANGUAGE MODELS

In accordance with ICLR 2026 policies, we disclose that a large language model (LLM) was used during paper preparation. Specifically, LLMs were employed for grammar correction, wording

refinement, and drafting of non-technical text passages. All research ideas, methods, experiments, and analyses were conceived, implemented, and validated by the authors, who take full responsibility for the correctness and integrity of the content.

