# OpenReview forum: "Play to Generalize: Learning to Reason Through Game Play"
_ICLR.cc/2026/Conference — ICLR 2026 Poster_

### Official Review · Reviewer_uv38 · 2025-10-20

**Soundness:** 3
**Presentation:** 3
**Contribution:** 2
**Rating:** 4
**Confidence:** 3

**Summary:**

The paper proposes Visual Game Learning (ViGaL), a post-training paradigm where an MLLM (7B) is fine-tuned via reinforcement learning on simple arcade-style games (e.g., Snake and a Rotation task) to acquire transferable reasoning skills. Without using any math/benchmark-specific data during RL, ViGaL improves downstream multimodal reasoning (MathVista and MMMU) while preserving general visual abilities, outperforming specialist models trained on curated reasoning data.

**Strengths:**

1. Overall, this is a clearly written and well-structured paper with polished figures that make the pipeline easy to follow from setup through results.

2. The core idea—leveraging controllable visual games with an RL loop as a scalable surrogate to induce transferable reasoning—is novel and timely, pushing beyond curated math datasets toward a more generalizable training signal.

3. Empirically, the paper shows that gameplay-driven RL improves downstream multimodal reasoning without eroding general vision capabilities, and it consistently outperforms SFT when trained on the same data. The evidence is strengthened by broad evaluations and careful ablations (modalities, difficulty control, data scaling), suggesting robustness rather than benchmark luck.

4. The approach is also practical: simple game environments, rule-based rewards, and a straightforward RL recipe lower the barrier to adoption and invite community exploration.

**Weaknesses:**

1. The paper proposes a promising direction, but the evidence base feels narrow: only two games (Snake and Rotation) are used, which makes it hard to argue for broad “play-to-generalize.” The paper explains that these two games let the authors probe “reasoning and perception” (Snake for spatial/planning, Rotation for angle perception) , yet this rationale is thin and the selection criteria appear ad hoc.

2. The idea of the paper would read as substantively stronger with diversity and robustness checks across more games.

3. The paper proposes that different games improve different math sub-skills, but the current analysis, while suggestive, is correlational and limited to two sources.

4. The paper proposes training via two games (Snake, Rotation) and shows encouraging transfer, but it does not yet establish games as a better alternative to standard post-training (e.g., curated math/logic data or instruction tuning) under controlled, budget-matched conditions.

**Questions:**

1. Can you articulate a principled game-selection taxonomy (e.g., planning, geometric transforms, partial observability, stochasticity) and explain why Snake and Rotation instantiate distinct buckets?

2. What ex-ante criteria (beyond convenience) determined these two games, and which candidate games were considered but rejected? If any rejected candidates, were they rejected before or after experiments, and why?

3. Do you observe negative transfer for any added game types?

---

> ### Author Response · Authors · 2025-11-23
> **Response to Reviewer uv38 (1/4)**
>
> We sincerely appreciate your constructive comments and your recognition of our paper's clarity and contribution. We address your concerns point by point below.
>
> ---
>
> > **W1**: The paper proposes a promising direction, but the evidence base feels narrow: only two games (Snake and Rotation) are used, which makes it hard to argue for broad "play-to-generalize." The paper explains that these two games let the authors probe "reasoning and perception" (Snake for spatial/planning, Rotation for angle perception), yet this rationale is thin and the selection criteria appear ad hoc.
>
> > **W2**: The idea of the paper would read as substantively stronger with diversity and robustness checks across more games.
>
> **Response to W1 and W2**: We appreciate this feedback. To address the concern about the narrow evidence base and to strengthen our claims about generalization from gameplay, we conducted an extended study with four additional games: Maze, Tetris, Sokoban, and Sudoku. Maze and Tetris were specifically chosen based on [Reviewer w1f6's suggestions](https://openreview.net/forum?id=u1tsgXPh2o&noteId=club8Gis6t) in Weakness 2, while Sokoban and Sudoku are representative games commonly used to evaluate LLMs in prior work [1,2,3]. Maze involves pathfinding through obstacles to reach a goal. Tetris requires rotating and positioning falling shapes to complete rows. Sokoban requires pushing boxes to target positions while navigating spatial constraints. Sudoku demands filling grids while satisfying row, column, and box constraints. These games introduce cognitive challenges beyond Snake and Rotation, including strategic planning and logical deduction.
>
> The table below presents results for all six games, comparing the four newly added games with our original two (Snake and Rotation):
>
> | Model | Math Avg. | MathVerse | MathVista | MathVision |
> |-------|-----------|-----------|-----------|------------|
> | Qwen2.5-VL-7B | 47.7 | 49.0 | 68.0 | 26.0 |
> | + Snake | 49.4 | 51.1 | 70.7 | 26.5 |
> | + Rotation | 49.3 | 50.4 | 71.2 | 26.3 |
> | + Maze (new) | 49.3 | 50.9 | 70.3 | 26.7 |
> | + Tetris (new) | 49.2 | 50.8 | 69.4 | 27.4 |
> | + Sokoban (new) | 49.3 | 50.5 | 70.1 | 27.2 |
> | + Sudoku (new) | 49.3 | 51.2 | 69.0 | 27.6 |
>
>
> These results show consistent improvements across all six games, with average gains of 1.5%-1.7% over the baseline. This stable pattern across mechanically diverse games supports our claim that multimodal reasoning can emerge from gameplay and that the transfer process generalizes beyond specific game mechanics rather than depending on our initial pair alone.
>
>
> ---

---

> > ### Author Response · Authors · 2025-11-23
> > **Response to Reviewer uv38 (2/4)**
> >
> > > **W3**: The paper proposes that different games improve different math sub-skills, but the current analysis, while suggestive, is correlational and limited to two sources.
> >
> > **Response to W3**: We appreciate this comment. Building on the extended experiments described in our response to W1 and W2, we conducted a detailed analysis of how different game mechanics transfer to specific mathematical sub-skills.
> >
> > We follow the official subject split used in MathVerse, which assigns each problem to one of three subjects: Plane Geometry (angles and lengths), Functions (coordinate graphs and related diagrams), and Solid Geometry (volume and surface area). We use this subject-level split instead of the finer subfields in Figure 3 of our manuscript because many of those subfields depend on overlapping skills, for example the Coordinate, Property, and Expression subfields all require interpreting coordinate systems. The subject-level grouping therefore offers a more stable and disentangled proxy for distinct mathematical reasoning abilities, for instance by placing all questions that require interpreting coordinate systems under Functions. We report performance gains for all six games separately for each subject.
> >
> > | Model | Plane Geometry | Functions | Solid Geometry |
> > |-------|----------|----------|----------|
> > | Baseline (Qwen2.5-VL-7B) | 52.43 | 49.81 | 33.45 |
> > | +Snake | +1.81 | +3.77 | +0.84 |
> > | +Rotation | +2.32 | +1.13 | +0.50 |
> > | +Maze (new) | +1.96 | +1.76 | +1.84 |
> > | +Tetris (new) | +1.73 | +2.01 | +1.34 |
> > | +Sokoban (new) | +1.49 | +1.89 | +0.84 |
> > | +Sudoku (new) | +2.28 | +2.27 | +1.34 |
> >
> > Based on these performance profiles, we grouped the six games into three clusters using K-Means clustering to identify systematic transfer patterns:
> >
> > - **Cluster 1 (Snake, Sokoban)**: Strongest transfer to **Functions** (+2.83% avg). Snake and Sokoban require tracking multiple objects moving in coordinate space: the snake's growing body segments or boxes being pushed to target positions. Players must reason about how coordinate relationships between objects evolve as actions unfold, mirroring the core skill in Functions problems, understanding coordinate transformations (e.g., translations, scaling) and how values change under sequential operations.
> > - **Cluster 2 (Rotation)**: Strongest transfer to **Plane Geometry** (+2.32%). This puzzle requires recognizing 3D object rotation angles, directly engaging angle reasoning and spatial relationships fundamental to plane geometry.
> > - **Cluster 3 (Maze, Sudoku, Tetris)**: Balanced improvements across all three subjects. These games engage distinct reasoning modes: Maze requires navigating to find optimal paths; Sudoku demands symbolic constraint satisfaction without spatial movement; Tetris involves shape transformations and pattern completion. This diversity of non-overlapping cognitive demands supports reasoning across mathematical domains more broadly rather than specializing to a single subject.
> >
> > These findings extend beyond initial correlational observations to reveal systematic relationships between game mechanics and mathematical reasoning categories. We have updated the manuscript accordingly: a new subsection (Appendix B.4: "Generalization Beyond Snake and Rotation: A Multi-Game Study") presents the complete analysis, and Section 4.1 now references this extended study.
> >
> > [1] Hu, Lanxiang, et al. "LMGame-Bench: How Good Are LLMs at Playing Games?" arXiv:2505.15146.
> >
> > [2] Paglieri, Davide, et al. "BALROG: Benchmarking Agentic LLM and VLM Reasoning on Games." arXiv:2411.13543.
> >
> > [3] Zhang, Haoran, et al. "ING-VP: MLLMs Cannot Play Easy Vision-Based Games Yet." arXiv:2410.06555.
> >
> > ---

---

> > > ### Author Response · Authors · 2025-11-23
> > > **Response to Reviewer uv38 (3/4)**
> > >
> > > > **W4**: The paper proposes training via two games (Snake, Rotation) and shows encouraging transfer, but it does not yet establish games as a better alternative to standard post-training (e.g., curated math/logic data or instruction tuning) under controlled, budget-matched conditions.
> > >
> > > **Response to W4**: Thank you for raising this important question. We address this concern by demonstrating two key findings:
> > >
> > > **Game-based post-training achieves competitive performance with strong data efficiency.** We conducted a comprehensive comparison against concurrent work on the same base model (Qwen2.5-VL-7B), comparing both math performance and dataset size. We adopt the same training recipes and training framework as in MM-Eureka-Qwen-7B so that this comparison is conducted under closely matched optimization conditions. As shown below, post-training on our game data achieves the highest math performance (50.6%) among all methods, while using only 72K training samples (36K per game for Snake and Rotation). This shows strong data efficiency, especially when compared to methods using standard curated data that require 136K to 270K samples yet reach lower performance (38.1% to 48.7%). MM-Eureka-Qwen-7B achieves comparable performance (50.1%) with fewer samples (12K), but their dataset consists entirely of manually curated multimodal mathematical reasoning problems with high human annotation costs, whereas our game data is purely synthetic and scalable without any human annotation.
> > >
> > > | Model | Math Avg. | Dataset Size | Math Dataset |  Game Dataset |
> > > |-------|-----------|--------------|------------------------------|-------------------|
> > > | R1-OneVision-7B | 46.8 | 155K | R1-OneVision Dataset [1] | - |
> > > | R1-VL-7B | 42.7 | 270K | Mulberry-260K [2], R1-VL-10K [3] | - |
> > > | Reason-RFT-Zero-7B | 38.1 | 136K | Reason-RFT-CoT [5] | - |
> > > | VLAA-Thinker-7B | 48.7 | 151K | VLAA-Thinking [6] | - |
> > > | OpenVLThinker-7B | 47.8 | 59K | OpenVLThinker-v12 [7] | - |
> > > | MM-Eureka-Qwen-7B | 50.1 | 12K | MMK12 [4] | - |
> > > | Ours | 50.6 | 72K | - | Snake and Rotation |
> > >
> > > **Game data complements standard curated math data.** Our game data is not meant to fully replace standard curated math data but rather to complement it. When we continue training on standard curated math data after game training, we obtain an additional 1.2% improvement on math benchmarks (from 50.6% to 51.8%). This shows that training on games builds core reasoning capabilities that further improve the effect of subsequent training on standard curated math data.
> > >
> > > | Model | Math Avg. | Dataset Size | Math Dataset |  Game Dataset |
> > > |-------|-----------|--------------|--------------|----------------|
> > > | MM-Eureka-Qwen-7B | 50.1 | 12K | MMK12 [4] | - |
> > > | Ours | 50.6 | 72K | - | Snake and Rotation |
> > > | Ours + Math Data | 51.8 (+1.2) | 12K + 72K | MMK12 [4] | Snake and Rotation |
> > >
> > >
> > > [1] Yang, Yi, et al. "R1-OneVision Dataset." Hugging Face, 2025. `https://huggingface.co/datasets/Fancy-MLLM/R1-Onevision`. Associated with arXiv:2503.10615.
> > >
> > > [2] Yao, Huanjin. "Mulberry-SFT." Hugging Face, 2025. `https://huggingface.co/datasets/HuanjinYao/Mulberry-SFT`.
> > >
> > > [3] Chen, Liang, et al. "R1-VL-10K." Hugging Face, 2025. `https://huggingface.co/datasets/jingyiZ00/R1-VL-10K`. Associated with R1-V project: `https://github.com/Deep-Agent/R1-V`.
> > >
> > > [4] Meng, Fanqing, et al. "MMK12." Hugging Face, 2025. `https://huggingface.co/datasets/FanqingM/MMK12`. Associated with arXiv:2503.07365.
> > >
> > > [5] Tan, Huajie, et al. "Reason-RFT-CoT-Dataset." Hugging Face, 2025. `https://huggingface.co/datasets/tanhuajie2001/Reason-RFT-CoT-Dataset`. Associated with arXiv:2503.20752.
> > >
> > > [6] Chen, Hardy, et al. "VLAA-Thinking." Hugging Face, 2025. `https://huggingface.co/datasets/UCSC-VLAA/VLAA-Thinking`. Associated with arXiv:2504.11468.
> > >
> > > [7] Deng, Yihe, et al. "OpenVLThinker-v12-Datasets." Hugging Face, 2025. `https://huggingface.co/collections/ydeng9/openvlthinker-v12-datasets`. Associated with arXiv:2503.17352.
> > >
> > > ---

---

> ### Author Response · Authors · 2025-11-23
> **Response to Reviewer uv38 (4/4)**
>
> > **Q1**: Can you provide a principled game-selection taxonomy (e.g., planning, geometric transforms, partial observability, stochasticity) and explain why Snake and Rotation instantiate distinct buckets?
>
> **Response to Q1**: Thank you for this question. To provide a principled taxonomy, we conducted a systematic study as detailed in our response to W3. We analyzed the performance improvements of six games on MathVerse across three subjects (Plane Geometry, Functions, Solid Geometry) and grouped them into three clusters using K-Means clustering based on their performance profiles. This data-driven analysis yielded three distinct clusters:
>
> - **Cluster 1 (Snake, Sokoban)**: Strongest transfer to **Functions** (+2.83% avg). Snake and Sokoban require tracking multiple objects moving in coordinate space: the snake's growing body segments or boxes being pushed to target positions. Players must reason about how coordinate relationships between objects evolve as actions unfold, mirroring the core skill in Functions problems, understanding coordinate transformations (e.g., translations, scaling) and how values change under sequential operations.
> - **Cluster 2 (Rotation)**: Strongest transfer to **Plane Geometry** (+2.32%). This puzzle requires recognizing 3D object rotation angles, directly engaging angle reasoning and spatial relationships fundamental to plane geometry.
> - **Cluster 3 (Maze, Sudoku, Tetris)**: Balanced improvements across all three subjects. These games engage distinct reasoning modes: Maze requires navigating to find optimal paths; Sudoku demands symbolic constraint satisfaction without spatial movement; Tetris involves shape transformations and pattern completion. This diversity of non-overlapping cognitive demands supports reasoning across mathematical domains more broadly rather than specializing to a single subject.
>
> Snake and Rotation fall into distinct clusters because they develop fundamentally different mathematical sub-skills. Rotation yields the strongest transfer to Plane Geometry (+2.32%), which involves angle-based problems and geometric transformations. This aligns with the game's core mechanic: determining which rotation (e.g., 90-degree rotation around the x-axis) aligns a 3D object with a target orientation, directly exercising the same angle reasoning required in plane geometry questions.
>
> In contrast, Snake provides the largest improvements on Functions (+3.77%), which involves coordinate systems and diagrams. This aligns with the game's requirement to track the snake's position and food locations in a 2D coordinate grid, where players must reason about coordinate relationships and how positions change through sequential moves.
>
> These distinct transfer patterns, angle-based geometry skills versus coordinate-based function skills, justify their placement in separate clusters and demonstrate that different games systematically develop different mathematical reasoning capabilities.
>
> ---
>
> > **Q2**: What ex-ante criteria (beyond convenience) determined these two games, and which candidate games were considered but rejected? If any rejected candidates, were they rejected before or after experiments, and why?
>
> **Response to Q2**: Thank you for your genuine question! We did not use a specialized selection process biased towards Snake and Rotation.
> We started with Snake for convenience after coming across the website for [SnakeBench](https://snakebench.com/). After observing positive signals with Snake, we then considered Rotation as a representative for perception ability.
> No other candidate games were discarded during the project. As described in our responses to W1, W2, and W3, we later added four more games (Maze, Tetris, Sokoban, and Sudoku) using the same training setup, and all of them also improved downstream reasoning. Our goal in this work is to show that multimodal reasoning can emerge from training on a broad range of game data, rather than relying on a carefully tuned game choice or hand-crafted selection criteria.
>
> However, we expect that extremely hard games, where the LLM cannot obtain any reward at the beginning, would be very difficult to train on and would likely make it hard to improve game performance or to generalize to other reasoning tasks.
>
> ---
>
> > **Q3**: Do you observe negative transfer for any added game types?
>
> **Response to Q3**: Yes. Although training on games can transfer to multimodal reasoning tasks and preserve general vision capability, we do observe negative transfer on OCR tasks. As shown in Table 9 in our manuscript, training on games slightly decreases OCR and Chart task performance (average drop from 83.3 to 82.2).

---

### Official Review · Reviewer_w1f6 · 2025-10-26

**Soundness:** 2
**Presentation:** 3
**Contribution:** 2
**Rating:** 4
**Confidence:** 4

**Summary:**

This paper proposes ViGaL (Visual Game Learning), a post-training method that applies reinforcement learning (RL) on simple visual games—primarily Snake—to improve the multimodal reasoning performance of a 7B MLLM on downstream benchmarks such as MathVista and MMMU. The authors show that models trained on these games achieve modest gains on reasoning tasks, even though no domain-specific reasoning data is used during training. The core claim is that gameplay can serve as a surrogate task to develop transferable reasoning skills.

While the empirical results demonstrate some performance improvement, the work largely repackages well-established ideas—such as multi-task learning, curriculum learning, and pretext tasks—under a new but superficially motivated paradigm. Crucially, it fails to provide any principled insight into why or when such transfer occurs, nor does it offer a methodology for designing effective auxiliary tasks for a given downstream goal. As such, the contribution appears incremental rather than foundational.

**Strengths:**

1) Clean Experimental Setup and Broad Benchmarking: The paper evaluates the proposed method on a wide range of established multimodal reasoning benchmarks (e.g., MathVista, MMMU, CLEVR+), which lends credibility to the reported performance gains. The experimental design is generally sound, and the use of rule-based RL avoids the complexity of reward modeling.

2) Demonstration of Cross-Domain Performance Gain: It is empirically shown that training on a simple game can lead to measurable improvements on unrelated reasoning tasks. This serves as a proof-of-concept that some form of transfer is possible, even if the mechanism remains opaque.

**Weaknesses:**

1) Lack of Novelty: Repackaging Known Ideas Without Substantial Advance

The idea that training on one task can improve performance on another—i.e., multi-task learning (MTL) or transfer learning—is decades old. The use of pretext tasks in self-supervised learning has been standard in vision and NLP. Even within RL, curriculum learning and autocurricula have long demonstrated that simple environments can give rise to complex behaviors. The paper does not convincingly argue why using Snake as a surrogate task is fundamentally different or more effective than prior approaches. It presents an anecdotal case rather than a general principle.

2) No Theoretical or Mechanistic Insight

What specific skills are learned in Snake that transfer to math reasoning?

Are there measurable properties of an auxiliary task (e.g., action space complexity, reward sparsity) that predict transfer success?

The most pressing practical question—how should one design or discover an effective auxiliary task for a given downstream goal?—is completely unaddressed. The choice of Snake appears arbitrary. Why not Tetris? Why not a maze? The paper provides no criteria, heuristics, or framework for answering this.

**Questions:**

1) Has the phenomenon of using simple, structured tasks to improve performance on complex ones been studied before the LLM era? If so, shouldn't this paper engage more deeply with that literature to justify its claimed novelty?

2) Please see the weakness.

---

> ### Author Response · Authors · 2025-11-23
> **Response to Reviewer w1f6 (1/4)**
>
> We sincerely appreciate your constructive feedback and the opportunity to address your concerns. We respond to each point below.
>
> ---
>
> > **W1**: Lack of Novelty: Repackaging Known Ideas Without Substantial Advance. The idea that training on one task can improve performance on another is decades old... Even within RL, curriculum learning has long shown that simple environments can lead to complex behaviors. The paper does not convincingly argue why using Snake as a surrogate task is different... It presents an anecdotal case rather than a general principle.
>
> **Response to W1**: We thank the reviewer for positioning our work within the broader history of transfer and curriculum learning. We agree that the *principle* of simple tasks leading to complex behaviors is established in pre-LLM literature [1, 2]. However, we respectfully argue that our work represents a distinct and non-trivial advance within the current MLLM landscape. We address each specific concern below:
>
> > The idea that training on one task can improve performance on another (i.e., multi-task learning or transfer learning) is decades old. The use of pretext tasks in self-supervised learning has been standard in vision and NLP.
>
> We do not claim to invent the concept of transfer learning; rather, we demonstrate that RL on games enables **reasoning process transfer** that is fundamentally different from standard representation learning. Our method transfers reasoning policies rather than data distributions. Standard transfer learning (e.g., ImageNet pre-training) relies on learning robust feature representations shared across domains. If our method were simply transferring features or "facts" about the game, Supervised Fine-Tuning (SFT) on the game data should yield improvements. However, our ablation study (Table 5f) reveals the opposite: SFT on game data degrades mathematical reasoning performance by **9.7%**, while RL on the same data yields a **12.3% improvement**. This indicates that the model benefits not from the *data distribution* but from the *optimization of the reasoning policy*. Through RL on games, the model learns reasoning strategies, including look-ahead planning, spatial verification, and constraint satisfaction, that generalize to mathematical tasks.
>
> > Even within RL, curriculum learning and autocurricula have long demonstrated that simple environments can give rise to complex behaviors.
>
> While we agree that prior curriculum learning in RL has shown that simple environments can give rise to complex behaviors, these works typically study improvement within a similar task (e.g., from simple games to complex games) [3]. In contrast, our results demonstrate a cross-task generalization step: from **Visual Control** (pixel-based survival games) to **Abstract Reasoning** (symbolic coordinate geometry and multi-discipline QA). This transfer occurs zero-shot, without the model ever observing worked solutions or equations during RL on game data. Our findings suggest that what transfers is the structural logic of the task. Skills learned in simple visual control games are reused when solving complex mathematical and multi-discipline reasoning problems. This means that, rather than only increasing behavioral complexity within the same task, training in simple visual games improves performance on harder reasoning benchmarks. This effect is qualitatively different from what has been reported in prior curriculum learning in RL.
>
> > The paper does not convincingly argue why using Snake as a surrogate task is fundamentally different or more effective than prior approaches. It presents an anecdotal case rather than a general principle.
>
> To address the concern that our results are anecdotal and specific to Snake, we demonstrate that the transfer effect generalizes across multiple games, not just Snake. Our extended study (detailed in Response to W2) confirms this pattern across four additional games (Maze, Tetris, Sokoban, and Sudoku), where all game-trained models yielded consistent reasoning gains, with average math improvements of 1.5%-1.7% over the baseline. This establishes a general principle: synthetic game environments serve as data-efficient proxies for developing complex reasoning behaviors. Unlike prior RL works focused on mastering specific environments (e.g., Atari agents), our objective is to use games to develop general reasoning capabilities. A generalist 7B model, post-trained *only* on synthetic games, can match or outperform specialist models trained on domain-specific math data, as shown in Table 2 in our manuscript.
>
> [1] Bengio, Yoshua, et al. "Curriculum Learning." ICML 2009.
>
> [2] Gidaris, Spyros, Praveer Singh, and Nikos Komodakis. "Unsupervised Representation Learning by Predicting Image Rotations." arXiv:1803.07728.
>
> [3] Narvekar, Sanmit, et al. "Curriculum learning for reinforcement learning domains: A framework and survey." Journal of Machine Learning Research 21.181 (2020): 1-50.
>
> ---

---

> > ### Author Response · Authors · 2025-11-23
> > **Response to Reviewer w1f6 (2/4)**
> >
> > > **W2**: No Theoretical or Mechanistic Insight. What specific skills are learned in Snake that transfer to math reasoning? Are there measurable properties of an auxiliary task (e.g., action space complexity, reward sparsity) that predict transfer success? The most pressing practical question, how should one design or discover an effective auxiliary task for a given downstream goal?, is completely unaddressed. The choice of Snake appears arbitrary. Why not Tetris? Why not a maze? The paper provides no criteria, heuristics, or framework for answering this.
> >
> > **Response to W2**: We appreciate this question. We agree that providing more mechanistic insight is important. Therefore, we conducted an extended study by post-training on four additional games: Maze, Tetris, Sokoban, and Sudoku. Maze and Tetris follow your suggestions, while Sokoban and Sudoku are representative games widely used in prior work on LLM evaluation [1,2,3]. Maze involves pathfinding through obstacles to reach a goal. Tetris requires rotating and positioning falling shapes to complete rows. Sokoban requires pushing boxes to target positions while navigating spatial constraints. Sudoku demands filling grids while satisfying row, column, and box constraints.
> >
> > The table below presents results for all six games, comparing the four newly added games with our original two (Snake and Rotation):
> >
> > **Table 1: Overall performance across three mathematical reasoning benchmarks**
> >
> > | Model | Math Avg. | MathVerse | MathVista | MathVision |
> > |-------|-----------|-----------|-----------|------------|
> > | Qwen2.5-VL-7B | 47.7 | 49.0 | 68.0 | 26.0 |
> > | + Snake | 49.4 | 51.1 | 70.7 | 26.5 |
> > | + Rotation | 49.3 | 50.4 | 71.2 | 26.3 |
> > | + Maze (new) | 49.3 | 50.9 | 70.3 | 26.7 |
> > | + Tetris (new) | 49.2 | 50.8 | 69.4 | 27.4 |
> > | + Sokoban (new) | 49.3 | 50.5 | 70.1 | 27.2 |
> > | + Sudoku (new) | 49.3 | 51.2 | 69.0 | 27.6 |
> >
> > To understand the mechanisms behind these improvements, we analyze subject-level performance on MathVerse. We follow the official subject split used in MathVerse, which assigns each problem to one of three subjects: Plane Geometry (angles and lengths), Functions (coordinate graphs and related diagrams), and Solid Geometry (volume and surface area). We use this subject-level split instead of the finer subfields in Figure 3 of our manuscript because many of those subfields depend on overlapping skills, for example the Coordinate, Property, and Expression subfields all require interpreting coordinate systems. The subject-level grouping therefore offers a more stable and disentangled proxy for distinct mathematical reasoning abilities, for instance, by placing all questions that require interpreting coordinate systems under Functions. Table 2 reports performance gains for all six games separately for each subject:
> >
> > **Table 2: Performance improvements by mathematical subject on MathVerse**
> >
> > | Model | Plane Geometry | Functions | Solid Geometry |
> > |-------|----------|----------|----------|
> > | Baseline (Qwen2.5-VL-7B) | 52.43 | 49.81 | 33.45 |
> > | +Snake | +1.81 | +3.77 | +0.84 |
> > | +Rotation | +2.32 | +1.13 | +0.50 |
> > | +Maze (new) | +1.96 | +1.76 | +1.84 |
> > | +Tetris (new) | +1.73 | +2.01 | +1.34 |
> > | +Sokoban (new) | +1.49 | +1.89 | +0.84 |
> > | +Sudoku (new) | +2.28 | +2.27 | +1.34 |

---

> ### Author Response · Authors · 2025-11-23
> **Response to Reviewer w1f6 (3/4)**
>
> (continue the response to W2)
>
> We now address each sub-question below.
>
> > The choice of Snake appears arbitrary. Why not Tetris? Why not a maze?
>
> The extended study above demonstrates that Snake is a representative game that can produce transferable reasoning skills. As shown in Table 1, all six games yielded consistent gains across MathVerse, MathVista, and MathVision, with average improvements of 1.5%-1.7%. Tetris and Maze, as you suggest, yield consistent improvements (1.5% and 1.6% average gains, respectively), confirming that the phenomenon is robust across different game mechanics. The choice of which specific game to use depends on the downstream task requirements, which we address below.
>
> > What specific skills are learned in Snake that transfer to math reasoning?
>
> Table 2 reveals that different games develop different transferable skills. Snake significantly improves performance on Functions problems involving coordinate systems and diagrams (+3.77%). As shown in Figure 4 in our manuscript, we explain this by comparing the reasoning processes required for playing Snake versus solving math problems. Solving Functions problems involves understanding coordinate transformations and how values change under sequential operations, which closely aligns with Snake's requirement to track multiple objects moving in coordinate space and reason about how coordinate relationships evolve. These results suggest that playing Snake develops fundamental skills like coordinate-based sequential reasoning that transfer to visual math questions.
>
> > Are there measurable properties of an auxiliary task that predict transfer success? How should one design or discover an effective auxiliary task for a given downstream goal?
>
> Based on the performance profiles in Table 2, we conducted an exploratory K-Means clustering on the performance profiles of the six games. We find a relation between game mechanics and transfer patterns, which suggests three tentative groups to guide auxiliary task design:
>
> - **To improve Functions (coordinate systems and diagrams):** Select games requiring *multi-object tracking in coordinate space* (e.g., Snake, Sokoban - Cluster 1). This specific alignment yields a +3.77% gain in Functions for Snake.
> - **To improve Plane Geometry (angles, lengths):** Select games requiring *spatial transformation and angle recognition* (e.g., Rotation - Cluster 2), which aligns with the +2.32% gain observed in geometry tasks.
> - **To improve General Reasoning:** Select games involving *path planning or symbolic constraints* (e.g., Maze, Sudoku - Cluster 3), which provide balanced improvements across diverse mathematical domains.
>
> This framework offers a concrete heuristic for discovery: effective auxiliary tasks can be identified by mapping their core game mechanics (e.g., grid navigation, rotation) to the cognitive demands of the target downstream task. We have incorporated this full analysis into the manuscript (Appendix B.4: "Generalization Beyond Snake and Rotation: A Multi-Game Study") and updated Section 4.1 to reference these design criteria. And we will continue to explore this direction in the future.
>
>
> [1] Hu, Lanxiang, et al. "LMGame-Bench: How Good Are LLMs at Playing Games?" arXiv:2505.15146.
>
> [2] Paglieri, Davide, et al. "BALROG: Benchmarking Agentic LLM and VLM Reasoning on Games." arXiv:2411.13543.
>
> [3] Zhang, Haoran, et al. "ING-VP: MLLMs Cannot Play Easy Vision-Based Games Yet." arXiv:2410.06555.
>
> ---

---

> ### Author Response · Authors · 2025-11-23
> **Response to Reviewer w1f6 (4/4)**
>
> > **Q1**: Has the phenomenon of using simple, structured tasks to improve performance on complex ones been studied before the LLM era? If so, shouldn't this paper engage more deeply with that literature to justify its claimed novelty?
>
> **Response to Q1**: Thank you for this question. Yes, the principle of simple tasks leading to complex behaviors is established in pre-LLM literature. As detailed in our **Response to W1**, we do not claim to invent transfer learning itself. Rather, we demonstrate that RL on games enables **reasoning process transfer** that is fundamentally different from standard transfer learning. Our method transfers reasoning policies rather than data distributions, as evidenced by our ablation study showing that SFT on game data degrades mathematical reasoning by 9.7%, while RL on the same data yields a 12.3% improvement. Furthermore, our work demonstrates cross-task generalization from visual control games to abstract mathematical reasoning, which is qualitatively different from prior curriculum learning that typically operates within similar tasks. In the revised manuscript, we have expanded the Related Work section to discuss transfer learning, self-supervised pretext tasks, and curriculum learning, and to clarify how our work differs from these lines of work.
>
> ---
>
> > **Q2**: Please see the weakness.
>
> **Response to Q2**: We have addressed the concerns raised in W1 and W2 above.
>
> ---

---

### Official Review · Reviewer_DBfm · 2025-10-29

**Soundness:** 3
**Presentation:** 3
**Contribution:** 3
**Rating:** 6
**Confidence:** 4

**Summary:**

The paper proposes using gameplay as a surrogate reasoning task to improve multimodal reasoning abilities of MLLM.
The model is post-trained with reinforcement learning on two simple games — Snake (2D grid planning) and Rotation (3D object rotation prediction). It uses a lightweight reward (accuracy + format) and no KL regularization. After training, the model shows improved performance on downstream reasoning benchmarks (math, geometry, and multimodal QA), while maintaining its visual perception capabilities.

**Strengths:**

- The idea of using structured games to indirectly train reasoning is novel and very interesting to me.
- The performance of using games improves  5–8% on math and spatial tasks, showing gameplay can transfer reasoning skills.
- The authors provided careful analysis on reward, prompts, and difficulty to show the effectiveness of each component.

**Weaknesses:**

- It is not very clear how to design a game for different types of reasoning abilities. That being said, although games are useful for post training, it requires design for each task.
- The authors are currently only designing two kinds of games with spatial/math-like games used, it is unclear whether other reasoning abilities can also be solved by games.
- The paper provides limited analysis of why the game can improve the math reasoning ability. Specifically, what kind of questions in the benchmark are these post-trained models most benefited from.

**Questions:**

Overall, this is a great paper. I would like to see the authors answers to my weaknesses.

---

> ### Author Response · Authors · 2025-11-23
> **Response to Reviewer DBfm (1/3)**
>
> We sincerely appreciate your positive feedback and acceptance recommendation. We address your remaining concerns below.
>
> ---
>
> > **W1**: It is not very clear how to design a game for different types of reasoning abilities. That being said, although games are useful for post training, it requires design for each task.
>
> **Response to W1**: We appreciate this concern about game design for different reasoning abilities. Our findings demonstrate that effective games for post-training already exist and can be systematically selected based on their cognitive requirements, eliminating the need to design new games for each task.
>
> **1. Diverse pre-existing games yield consistent reasoning improvements without requiring custom design.**
>
> We leverage existing games from established benchmarks [1, 2, 3] that were originally created to evaluate vision-language model capabilities. To validate this approach, we conducted experiments with four additional games beyond our original Snake and Rotation setup: Maze, Tetris, Sokoban, and Sudoku. Maze involves pathfinding through obstacles to reach a goal. Tetris requires rotating and positioning falling shapes to complete rows. Sokoban requires pushing boxes to target positions while navigating spatial constraints. Sudoku demands filling grids while satisfying row, column, and box constraints. These games introduce cognitive challenges beyond Snake and Rotation. All games showed consistent improvements on mathematical reasoning benchmarks:
>
> | Model | Math Avg. | MathVerse | MathVista | MathVision |
> |-------|-----------|-----------|-----------|------------|
> | Qwen2.5-VL-7B | 47.7 | 49.0 | 68.0 | 26.0 |
> | + Snake | 49.4 | 51.1 | 70.7 | 26.5 |
> | + Rotation | 49.3 | 50.4 | 71.2 | 26.3 |
> | + Maze (new) | 49.3 | 50.9 | 70.3 | 26.7 |
> | + Tetris (new) | 49.2 | 50.8 | 69.4 | 27.4 |
> | + Sokoban (new) | 49.3 | 50.5 | 70.1 | 27.2 |
> | + Sudoku (new) | 49.3 | 51.2 | 69.0 | 27.6 |
>
> These results demonstrate consistent improvements across all six games, with average gains of 1.5%-1.7% over the baseline. This pattern across mechanically diverse games supports our claim that multimodal reasoning can emerge from gameplay, generalizing beyond our initially chosen pair.
>
> **2. Clustering analysis reveals systematic mappings from game characteristics to target reasoning abilities.**
>
> We evaluated the performance improvements on MathVerse's three built-in subjects. We follow the official subject split used in MathVerse, which assigns each problem to one of three subjects: Plane Geometry (angles and lengths), Functions (coordinate graphs and related diagrams), and Solid Geometry (volume and surface area).
>
> | Model | Plane Geometry | Functions | Solid Geometry |
> |-------|----------|----------|----------|
> | Baseline (Qwen2.5-VL-7B) | 52.43 | 49.81 | 33.45 |
> | +Snake | +1.81 | +3.77 | +0.84 |
> | +Rotation | +2.32 | +1.13 | +0.50 |
> | +Maze (new) | +1.96 | +1.76 | +1.84 |
> | +Tetris (new) | +1.73 | +2.01 | +1.34 |
> | +Sokoban (new) | +1.49 | +1.89 | +0.84 |
> | +Sudoku (new) | +2.28 | +2.27 | +1.34 |
>
> Based on these performance profiles, we grouped the six games into three clusters using K-Means clustering to identify systematic transfer patterns:
>
> - **Cluster 1 (Snake, Sokoban)**: Strongest transfer to **Functions** (+2.83% avg). Snake and Sokoban require tracking multiple objects moving in coordinate space: the snake's growing body segments or boxes being pushed to target positions. Players must reason about how coordinate relationships between objects evolve as actions unfold, mirroring the core skill in Functions problems, understanding coordinate transformations (e.g., translations, scaling) and how values change under sequential operations.
> - **Cluster 2 (Rotation)**: Strongest transfer to **Plane Geometry** (+2.32%). This puzzle requires recognizing 3D object rotation angles, directly engaging angle reasoning and spatial relationships fundamental to plane geometry.
> - **Cluster 3 (Maze, Sudoku, Tetris)**: Balanced improvements across all three subjects. These games engage distinct reasoning modes: Maze requires navigating to find optimal paths; Sudoku demands symbolic constraint satisfaction without spatial movement; Tetris involves shape transformations and pattern completion. This diversity of non-overlapping cognitive demands supports reasoning across mathematical domains more broadly rather than specializing to a single subject.

---

> ### Author Response · Authors · 2025-11-23
> **Response to Reviewer DBfm (2/3)**
>
> (continue the response to W1)
>
> **3. These patterns provide clear guidance for game selection:**
>
> - **To improve Functions (coordinate systems and diagrams):** Select games requiring *multi-object tracking in coordinate space* (e.g., Snake, Sokoban - Cluster 1). This specific alignment yields a +3.77% gain in Functions for Snake.
> - **To improve Plane Geometry (angles, lengths):** Select games requiring *spatial transformation and angle recognition* (e.g., Rotation - Cluster 2), which aligns with the +2.32% gain observed in geometry tasks.
> - **To improve General Reasoning:** Select games involving *path planning or symbolic constraints* (e.g., Maze, Sudoku - Cluster 3), which provide balanced improvements across diverse mathematical domains.
>
> Practitioners can simply select from existing game repositories based on their target reasoning abilities. We have documented this framework in detail in Appendix B.4: "Generalization Beyond Snake and Rotation: A Multi-Game Study" and updated Section 4.1 to provide practical guidance for game selection.
>
> [1] Hu, Lanxiang, et al. "LMGame-Bench: How Good Are LLMs at Playing Games?" arXiv:2505.15146.
>
> [2] Paglieri, Davide, et al. "BALROG: Benchmarking Agentic LLM and VLM Reasoning on Games." arXiv:2411.13543.
>
> [3] Zhang, Haoran, et al. "ING-VP: MLLMs Cannot Play Easy Vision-Based Games Yet." arXiv:2410.06555.
>
> ---
>
> > **W2**: The authors are currently only designing two kinds of games with spatial and math-like games used. It is unclear whether other reasoning abilities can also be solved by games.
>
> **Response to W2**: We appreciate this concern and address both aspects of the question below.
>
> > The authors are currently only designing two kinds of games with spatial and math-like games used.
>
> To demonstrate that our approach is not limited to two specific games, we conducted extended experiments with four additional games (Maze, Tetris, Sokoban, and Sudoku) that target different cognitive abilities, as detailed in response to W1. These results demonstrate consistent improvements across all six games, with average gains of 1.5%-1.7% over the baseline. This stable pattern across mechanically diverse games indicates that the transfer process generalizes beyond specific game mechanics.
>
> > It is unclear whether other reasoning abilities can also be solved by games.
>
>  We evaluated our approach on MMMU and MMMU-Pro benchmarks in Table 3 of our manuscript. These benchmarks test college-level multi-disciplinary reasoning across six diverse fields: Art & Design, Business, Science, Health & Medicine, Humanities & Social Science, and Tech & Engineering. As shown in the table below, our game-trained models achieve significant improvements compared to other multimodal reasoning models post-trained on the same base model:
>
> | Model | Multi-Discipline Avg. | MMMU (val) | MMMU-Pro (overall) |
> |-------|----------------------|------------|-------------------|
> | Qwen2.5-VL-7B (baseline) | 45.7 | 54.3 | 37.0 |
> | R1-Onevision-7B | 42.3 | 51.9 | 32.6 |
> | R1-VL-7B | 39.7 | 50.0 | 29.4 |
> | MM-Eureka-Qwen-7B | 46.4 | 55.8 | 36.9 |
> | Reason-RFT-Zero-7B | 40.9 | 51.2 | 30.6 |
> | VLAA-Thinker-7B | 40.1 | 48.2 | 31.9 |
> | OpenVLThinker-7B | 38.5 | 54.8 | 22.1 |
> | ViGaL Snake | 46.2 | 55.8 | 36.6 |
> | ViGaL Rotation | 45.9 | 54.1 | 37.7 |
> | ViGaL Snake + Rotation | **47.7** | **58.0** | **37.4** |
>
> Our Snake + Rotation model achieves 58.0% on MMMU and 47.7% average on multi-discipline reasoning, outperforming all other post-training methods. Notably, it surpasses R1-OneVision-7B by 6.1% on MMMU and MM-Eureka-Qwen-7B by 1.3% on average, despite these methods being trained on educational or reasoning-specific data. This demonstrates that game-based training develops reasoning capabilities that transfer effectively to diverse academic disciplines beyond spatial and mathematical tasks.
>
> ---

---

> > ### Author Response · Authors · 2025-11-23
> > **Response to Reviewer DBfm (3/3)**
> >
> > > **W3**: The paper provides limited analysis of why the game can improve the math reasoning ability. Specifically, what kind of questions in the benchmark are these post-trained models most benefited from.
> >
> > **Response to W3**: We appreciate this comment about understanding why games improve math reasoning and which question types benefit most. We provide a comprehensive analysis of these transfer patterns.
> >
> > 1. **Question types benefiting most from Snake and Rotation**
> >
> > As shown in Figure 3 of our manuscript, we analyze per-category gains on MathVerse to understand which mathematical skills each game enhances. To directly answer what kind of questions benefit most from game-based post-training, we provide a detailed breakdown of subject-level performance on MathVerse. We follow the official subject split used in MathVerse, which assigns each problem to one of three subjects: Plane Geometry (angles and lengths), Functions (coordinate graphs and related diagrams), and Solid Geometry (volume and surface area). We use this subject-level split instead of the finer subfields in Figure 3 because many of those subfields depend on overlapping skills, for example the Coordinate, Property, and Expression subfields all require interpreting coordinate systems. The subject-level grouping therefore offers a more stable and disentangled proxy for distinct mathematical reasoning abilities, for instance, by placing all questions that require interpreting coordinate systems under Functions.
> >
> > The table below is the result and corresponding findings:
> >
> > | Subject | Snake | Rotation | Key Finding |
> > |---------|--------|----------|-------------|
> > | Functions | +3.77% | +1.13% | Snake excels at coordinate systems and diagrams |
> > | Plane Geometry | +1.81% | +2.32% | Rotation is stronger for angles and lengths |
> > | Solid Geometry | +0.84% | +0.50% | Both games show smaller gains here |
> >
> > Figure 4 in our manuscript illustrates why these patterns emerge by comparing game mechanics to mathematical problem-solving processes. For example:
> > - **Snake** requires tracking multiple objects moving in coordinate space (e.g., the snake's growing body segments), where players must reason about how coordinate relationships evolve as actions unfold, directly transferring to **Functions** (coordinate systems and diagrams).
> > - **Rotation** requires recognizing 3D object rotation angles, explaining its superior performance on **Plane Geometry** (angles and lengths).
> >
> > 2. **Question types benefiting most from all six games**
> >
> > Extending this analysis to all six games (as described in W1), the subject-level performance breakdown on MathVerse is shown below:
> >
> > | Model | Plane Geometry | Functions | Solid Geometry |
> > |-------|----------|----------|----------|
> > | Baseline (Qwen2.5-VL-7B) | 52.43 | 49.81 | 33.45 |
> > | +Snake | +1.81 | +3.77 | +0.84 |
> > | +Rotation | +2.32 | +1.13 | +0.50 |
> > | +Maze (new) | +1.96 | +1.76 | +1.84 |
> > | +Tetris (new) | +1.73 | +2.01 | +1.34 |
> > | +Sokoban (new) | +1.49 | +1.89 | +0.84 |
> > | +Sudoku (new) | +2.28 | +2.27 | +1.34 |
> >
> > Based on these performance profiles, we grouped the six games into three clusters using K-Means clustering to identify systematic transfer patterns:
> >
> > - **Cluster 1 (Snake, Sokoban)**: Strongest transfer to **Functions** (+2.83% avg). Snake and Sokoban require tracking multiple objects moving in coordinate space: the snake's growing body segments or boxes being pushed to target positions. Players must reason about how coordinate relationships between objects evolve as actions unfold, mirroring the core skill in Functions problems, understanding coordinate transformations (e.g., translations, scaling) and how values change under sequential operations.
> > - **Cluster 2 (Rotation)**: Strongest transfer to **Plane Geometry** (+2.32%). This puzzle requires recognizing 3D object rotation angles, directly engaging angle reasoning and spatial relationships that are central to plane geometry.
> > - **Cluster 3 (Maze, Sudoku, Tetris)**: Balanced improvements across all three subjects. These games engage distinct reasoning modes: Maze requires navigating to find optimal paths, Sudoku demands symbolic constraint satisfaction without spatial movement, and Tetris involves shape transformations and pattern completion. This diversity of non-overlapping cognitive demands supports reasoning across mathematical domains more broadly rather than specializing to a single subject.
> >
> > These findings demonstrate that specific game mechanics predictably enhance corresponding mathematical reasoning skills, providing clear guidance for selecting games to target particular weaknesses in mathematical understanding.
> >
> > ---

---

### Official Review · Reviewer_ZnCJ · 2025-11-01

**Soundness:** 3
**Presentation:** 3
**Contribution:** 3
**Rating:** 8
**Confidence:** 3

**Summary:**

The paper proposes Visual Game Learning (ViGaL)  that uses RL post-training on simple visual games (e.g., Snake and a Rotation task) to elicit reasoning capability in multimodal LLMs in other domains such as math. Instead of training on math or benchmark-style reasoning data, a MLLM is fine-tuned to play these games with rule-based rewards; the resulting model then transfers the acquired skills to out-of-domain tasks while retaining general visual abilities. The paper argues that gameplay can serve as a scalable surrogate task for RL post-training to unlock broadly useful multimodal reasoning.

**Strengths:**

- The paper shows that RL post-training purely on simple visual games (Snake, Rotation) yields measurable gains on other seemingly unrelated domains such as math despite no direct supervision from those tasks. This is a surprising finding and is worth spreading.
- The gameplay setup enables verifiable rule-based rewards that are friendly to reasoning training, avoiding the need for expensive reward models or human labels. The fact that it can generalize to other domains shows that it has high potential.
- Another benefit of using game play for reasoning training is that there are a large amount of games out there with highly diverse contents. They cover all kinds of aspects of human skills, and can be a rich supplement to reasoning training data.
- The fine-grained analysis in section 3.1 is interesting. For example it connects specific games to specific math subfields, e.g. the game Rotation aligns with angle/length questions.

**Weaknesses:**

* The games are relatively simple, which is understandable though because it is the first effort to explore this direction.

**Questions:**

* The prompts include thinking instructions synthesized by GPT-4o. Are they necessary? How much do they affect the results?

---

> ### Author Response · Authors · 2025-11-23
>
> We sincerely appreciate your positive feedback and acceptance recommendation.
>
> ---
>
> > **W1**: The games are relatively simple, which is understandable though because it is the first effort to explore this direction.
>
> **Response to W1**: We appreciate your understanding. While our initial focus was on Snake and Rotation, we have already validated that our findings generalize across mechanically diverse games. To address concerns about game diversity, we conducted extended experiments with four additional games: Maze, Tetris, Sokoban, and Sudoku. Maze involves pathfinding through obstacles to reach a goal. Tetris requires rotating and positioning falling shapes to complete rows. Sokoban requires pushing boxes to target positions while navigating spatial constraints. Sudoku demands filling grids while satisfying row, column, and box constraints. All six games showed consistent improvements on mathematical reasoning benchmarks:
>
> | Model | Math Avg. | MathVerse | MathVista | MathVision |
> |-------|-----------|-----------|-----------|------------|
> | Qwen2.5-VL-7B | 47.7 | 49.0 | 68.0 | 26.0 |
> | + Snake | 49.4 | 51.1 | 70.7 | 26.5 |
> | + Rotation | 49.3 | 50.4 | 71.2 | 26.3 |
> | + Maze (new) | 49.3 | 50.9 | 70.3 | 26.7 |
> | + Tetris (new) | 49.2 | 50.8 | 69.4 | 27.4 |
> | + Sokoban (new) | 49.3 | 50.5 | 70.1 | 27.2 |
> | + Sudoku (new) | 49.3 | 51.2 | 69.0 | 27.6 |
>
> These results demonstrate consistent improvements across all six games, with average gains of 1.5%-1.7% over the baseline. This stable pattern across mechanically diverse games confirms that multimodal reasoning can emerge from gameplay, and that the transfer process generalizes beyond our initial pair of games. We have added these additional results to Appendix B.4: "Generalization Beyond Snake and Rotation: A Multi-Game Study."
>
> ---
>
> > **Q1**: The prompts include thinking instructions synthesized by GPT-4o. Are they necessary? How much do they affect the results?
>
> **Response to Q1**: We appreciate this question about the necessity of GPT-4o synthesized thinking instructions. Our ablation studies provide clear insights into their impact and when they are required.
>
> The thinking instructions guide the model's reasoning process by providing specific problem-solving strategies, such as "finding the nearest apple by calculating Manhattan distances" for Snake. Our ablation study, as shown in Table 5a in our manuscript, indicates that these instructions yield significant improvements:
>
> | Prompt Design | Avg. | Math | CLEVR+ | Geometry |
> |---------------|------|------|---------|----------|
> | Base model | 49.1 | 47.7 | 54.9 | 44.8 |
> | w/o reasoning instructions | 59.5 | 48.0 | 80.4 | 50.1 |
> | **w/ reasoning instructions** | **62.3** | **49.4** | **82.6** | **55.0** |
>
> The reasoning instructions provide a 2.8% improvement in average accuracy (from 59.5% to 62.3%) across downstream benchmarks. These instructions are particularly necessary for complex games like Snake and Rotation, making them difficult to learn without guided reasoning strategies.
>
> However, we also validated that simpler games can improve performance without GPT-4o synthesized instructions. In our extended experiments with four additional games (Maze, Tetris, Sokoban, and Sudoku), the prompts contained only the game rules and objectives without synthesized reasoning strategies. Despite this simplified setup, all four games still achieved consistent improvements of 1.5%-1.7% on mathematical reasoning benchmarks (see our response to W1 for detailed results).
>
> This suggests a clear pattern: GPT-4o synthesized reasoning instructions are necessary for complex games with sparse rewards and multi-step planning requirements, but simpler games that are easier to learn through post-training can achieve meaningful improvements without them. This indicates that game complexity is a primary determinant for the necessity of instructions.

---

### Author Response · Authors · 2025-11-23
**General Response**

**Dear Reviewers, ACs, and SACs,**

We appreciate the constructive comments from all reviewers on our submission.

---

We are grateful that reviewers recognize this work as a **promising approach to training multimodal reasoning with simple visual games** and a **well-presented study with clear experiments and analysis**. Our work tests whether **RL post-training purely on simple, rule-based games such as Snake and Rotation can improve math and multimodal reasoning without using task-specific reasoning data**, using a **practical training pipeline with broad evaluations and ablations**.

---

We are encouraged by the reviewers' positive feedback, which highlights:

- **Game-based RL as a surrogate for reasoning**: Training on simple, structured visual games with rule-based rewards serves as a surrogate task for multimodal reasoning, avoiding expensive reward models or human labels (Reviewers `uv38`, `DBfm`, `ZnCJ`).

- **Measurable transfer to downstream tasks**: RL post-training only on Snake and Rotation, without math or benchmark-style data, yields gains on math and multimodal reasoning benchmarks while retaining visual ability (Reviewers `uv38`, `DBfm`, `ZnCJ`, `w1f6`).

- **Clean and practical training setup**: The pipeline uses simple game environments, lightweight and verifiable rewards, and an RL recipe that is easy to implement and extend (Reviewers `uv38`, `DBfm`, `w1f6`).

- **Broad benchmarking and analysis**: Results on several established benchmarks with ablations on reward design, prompts, and difficulty control (Reviewers `uv38`, `DBfm`, `w1f6`).

- **Fine-grained analysis of transfer patterns**: Analyses connecting game mechanics, such as grid planning and angle perception, to mathematical skills and subfields (Reviewers `ZnCJ`, `DBfm`, `uv38`).

---

To address reviewers' concerns, we added targeted experiments and analyses:

- **Extending beyond two games to a six-game study**: We added Maze, Tetris, Sokoban, and Sudoku on top of Snake and Rotation, all trained under the same protocol with a small budget per game. Across MathVerse, MathVista, and MathVision, every game consistently outperforms the base model, addressing concerns about narrow evidence and showing the effect is not tied to our original game pair (Reviewers `uv38`, `w1f6`, `DBfm`, `ZnCJ`).

- **Mechanistic analysis via subject-level gains and clustering**: Using MathVerse's subject split (Plane Geometry, Functions, Solid Geometry), we show that different games yield distinct gain profiles and can be grouped into clusters (Snake/Sokoban, Rotation, Maze/Sudoku/Tetris). This supports links between game mechanics (grid-based planning, angle and rotation perception, and symbolic constraints) and the mathematical skills they improve, responding to calls for mechanistic insight and principled auxiliary task design (Reviewers `uv38`, `w1f6`, `DBfm`).

- **Data efficiency and comparison to post-training on math**: We compare post-training on games versus math on the same base model (Qwen2.5-VL-7B). Our game-trained model achieves the strongest math performance among compared baselines while using far fewer synthetic game samples, and joint training with a strong math dataset yields further gains. These results show game-based post-training can be competitive with standard post-training under matched training budgets (Reviewer `uv38`).

- **Clarifying the role of synthesized reasoning instructions**: We report ablations on GPT-4o-generated thinking instructions for complex games like Snake and Rotation, showing that such instructions bring extra gains, while simpler games in our extended study improve reasoning even without them, clarifying when such guidance is most helpful (Reviewer `ZnCJ`).

- **Characterizing negative transfer**: We analyze domains where game training slightly hurts performance, finding a small but consistent drop on OCR and chart tasks, which answers the question on negative transfer and clarifies the trade-offs of game-based post-training (Reviewer `uv38`).

---

**Summary of revisions:**

- **New multi-game study and analysis** in `Appendix B.4` titled *"Generalization Beyond Snake and Rotation: A Multi-Game Study"*, adding experiments on Maze, Tetris, Sokoban, and Sudoku with subject-level transfer patterns and K-Means clustering over six games.

- **Updated main text discussion** in `Section 4.1` to reference the new multi-game study and to provide guidance for selecting games based on their mechanics and the target reasoning skills, in response to questions about principled game selection.

- **Expanded related work discussion** in `Section 2` Related Work to connect ViGaL to prior work on transfer learning, self-supervised pretext tasks, and curriculum learning, and to clarify how our contribution differs from these lines of research.

All revisions in the paper are highlighted in the updated manuscript. We sincerely thank the reviewers for their constructive feedback and the chance to improve our work.

---

### Meta-Review · Area_Chair_oELV · 2026-01-01

**Summary:**

The paper explores gameplay as a potential post-training approach for improving the generalization performance of large language models. Specifically, it claims that post-training an LLM using reinforcement learning, where rewards are derived from playing specific games, enables the model not only to excel at those games but also to generalize to playing other unrelated games, as well as to perform better on math and visual reasoning tasks that are not directly related to the games. The paper bases its experiments on two popular arcade games: Snake and Rotation. Experimental results show that the proposed approach, called ViGaL, outperforms QwenVL on gameplay generalization and yields 1–3% improvements in accuracy on math and visual reasoning benchmarks.

**AC Comments:**
The paper received mixed to positive reviews, with one accept, one borderline accept, and two borderline rejects. The key concerns raised in the initial reviews include: i) a lack of insight into why gameplay leads to improved performance; ii) the limited number of games (two) used in the original submission; iii) insufficient analysis of the game skills and abilities required to solve downstream tasks; iv) a lack of guidance on how to select games given specific downstream tasks; and v) missing comparisons between gameplay-based post-training and standard post-training approaches.

The authors provide a strong rebuttal with additional results that bring clarity and insight into several of these critical questions. New experiments incorporating additional games and a cluster analysis of skill sets offer valuable insight into why certain games help with specific problem types. Comparisons between standard post-training methods and gameplay-based post-training further help contextualize the empirical gains of the proposed approach. However, some open questions remain insufficiently addressed, including: i) the lack of a recipe for selecting games that could guarantee improvements; ii) whether gameplay is the most appropriate post-training paradigm, or whether alternative (e.g., self-supervised) approaches could better capture the relevant skill sets; and iii) whether the computational cost of gameplay-based post-training is justified relative to standard approaches, given the modest gains (+1–3%).

The authors are encouraged to further consider these questions. Nonetheless, the paper presents an interesting post-training methodology, and AC believes that the rebuttal experiments resolve many of the key concerns raised by the reviewers. As such, the paper appears to meet the bar for acceptance, and AC recommends acceptance.

**Reviewer Concerns:**

*Reviewer ZnCJ* overall liked the approach presented in the paper and considers the idea worth exploring further. The reviewer raises a minor concern regarding the relative simplicity of the games considered.

*Reviewer DBfm* raises an important concern about the lack of guidance on how to design or select games that could be beneficial, and whether different types of games might impart different skill sets. The reviewer also seeks clarity on the broader question of why gameplay improves mathematical reasoning, which is not adequately explained in the paper.

*Reviewer w1f6* raises several critical concerns: i) a lack of novelty in the proposed approach, which could be viewed as another form of multi-task or transfer learning, and a lack of insight into why games such as Snake lead to performance gains; ii) the absence of mechanistic explanations for the observed improvements; and iii) unclear guidance on how to design or discover effective game tasks for specific downstream objectives.

*Reviewer uv38* reiterates several concerns raised by other reviewers, including: i) the use of only two games in the RL setup; ii) the lack of analysis on correlations between the games and the skills learned; and iii) insufficient evidence that gameplay-based post-training outperforms standard post-training methods.

**Reviewer Scores:**

**Reviewer ZnCJ:** In response to concerns about the simplicity of the games considered, the authors report new results in the rebuttal involving additional games (Maze, Tetris, Sokoban, and Sudoku). Post-training on these games and evaluating on math benchmarks still yields modest improvements of 1–2%.

**Reviewer DBfm:** The authors present a clustering analysis that highlights relationships between game characteristics and specific reasoning abilities in downstream tasks. The analysis shows that Snake and Sokoban are mainly correlated with coordinate transformations; the Rotation game with plane geometry; and Maze, Sudoku, and Tetris with more general reasoning across problem types.

[*AC's thoughts on the response*]
The clustering analysis, together with the new results on additional games, provides important insights into how gameplay may be improving model performance. AC believes these explanations satisfactorily address the reviewer’s concerns.

**Reviewer w1f6:** In response to questions about transfer learning, the authors argue that supervised fine-tuning for gameplay leads to an approximately 10% drop in math reasoning performance, whereas reinforcement learning yields a 12% improvement. The authors further claim that gameplay enables the model to learn structural logic underlying the tasks, which in turn supports better generalization. However, the authors do not fully address the reviewer’s most pertinent question regarding how to select games that are effective for improving downstream task performance.

[*AC's thoughts on the response*]
The authors provide a thoughtful response that partially addresses the reviewer’s criticisms. However, the key question of how to design or select games for improving performance on complex downstream tasks remains unanswered. It is also worth considering whether explicitly designing games is necessary, or whether self-supervised objectives that encode desired problem-solving skills might suffice. Given the relatively modest generalization gains (1–3%), it is also unclear whether the computational cost of training models to play arbitrarily chosen games is justified. In light of these unresolved issues, AC believes the reviewer may not have been fully satisfied and would likely have maintained a score of borderline reject.

**Reviewer uv38:** The authors provide additional results with newly considered games and a correlation analysis, which AC believes sufficiently address the reviewer’s concerns. To demonstrate the benefits of gameplay-based post-training relative to standard approaches, the authors also present two additional results: i) training MM-Eureka-Qwen-7B using 12k curated data and 72k synthetic game data, which yields a 0.5% improvement; and ii) combining the 12k curated data with the 72k game data, resulting in a 1.7% improvement. These results support the empirical benefits of the gameplay-based approach.

[*AC's thoughts on the response*]
The responses are strong, and AC believes they adequately address the reviewer’s concerns.

---

### Decision · Program_Chairs · 2026-01-26

Accept (Poster)